# RISK-SENSITIVE VARIATIONAL ACTOR-CRITIC: A MODEL-BASED APPROACH

**Alonso Granados**
Department of Computer Science
University of Arizona
Tucson, AZ, USA
alonsog@cs.arizona.edu

**Reza Ebrahimi**
School of Information Systems and Management
University of South Florida
Tampa, FL, USA
ebrahimim@usf.edu

**Jason Pacheco**
Department of Computer Science
University of Arizona
Tucson, AZ, USA
pachecoj@cs.arizona.edu

## ABSTRACT

Risk-sensitive reinforcement learning (RL) with an entropic risk measure typically requires knowledge of the transition kernel or performs unstable updates w.r.t. exponential Bellman equations. As a consequence, algorithms that optimize this objective have been restricted to tabular or low-dimensional continuous environments. In this work we leverage the connection between the entropic risk measure and the RL-as-inference framework to develop a risk-sensitive variational actor-critic algorithm (rsVAC). Our work extends the variational framework to incorporate stochastic rewards and proposes a variational model-based actor-critic approach that modulates policy risk via a risk parameter. We consider, both, the risk-seeking and risk-averse regimes and present rsVAC learning variants for each setting. Our experiments demonstrate that this approach produces risk-sensitive policies and yields improvements in both tabular and risk-aware variants of complex continuous control tasks in MuJoCo.

## 1 INTRODUCTION

Deep reinforcement learning (RL) algorithms have contributed to many breakthroughs in domains such as games (Mnih et al., 2015) and robotics (Levine et al., 2016). However, the standard objective in RL, maximization of the expected sum of rewards, disregards the variability of the return due to the intrinsic uncertainty in the transition dynamics and the stochasticity of the rewards which can lead to catastrophic behavior. Such catastrophic behavior is especially common in real-world applications, such as autonomous driving agents acting dangerously to achieve high reward (Chia et al., 2022) or financial losses in portfolio management (Lai et al., 2011). As a consequence, risk-aware agents are important to adapt to inherent environmental risk.

Many risk measures have been studied to introduce risk-sensitivity into RL algorithms. For instance, value at risk (VaR) (Chow et al., 2018), conditional value at risk (CVaR) (Chow & Ghavamzadeh, 2014; Greenberg et al., 2022), mean-variance (Tamar et al., 2012; La & Ghavamzadeh, 2013), reward-volatility risk measure (Zhang et al., 2021) and Gini-deviation (Luo et al., 2024). In this work we focus on the entropic risk measure, an approach that incorporates risk into its objective via the exponential utility function (Howard & Matheson, 1972; Borkar, 2002). Directly optimizing this objective is challenging: it requires the knowledge of the transition kernel or it depends on unstable updates w.r.t. exponential Bellman equations (Noorani et al., 2023).

We address these challenges by exploiting the connection between RL and probabilistic inference to obtain a surrogate objective to the entropic risk measure. RL-as-inference algorithms search for policies that maximize the probability of optimal trajectories rather than maximizing the expected return. However, it has been observed that this objective can produce unwanted risk-seeking behaviour

in the learned policy (Levine, 2018; O'Donoghue et al., 2019; Tarbouriech et al., 2023). Many such methods take a variational approach that constrain the posterior dynamics to equal those of the true environment (Haarnoja et al., 2017; 2018), but can lead to overly stochastic policies (Fellows et al., 2019). Existing variational model-based methods allow posterior dynamics to vary (Chow et al., 2018) but lead to risk-seeking policies that do not adapt to aleatoric risk in the environment (Eysenbach et al., 2022). Furthermore, these methods assume a deterministic reward model that implicitly ignores its risk contribution to the original objective.

In this work, we leverage the connection between RL and probabilistic inference to formulate a variational lower bound on the entropic risk measure that can be optimized using only experience from an agent. We optimize this surrogate objective using an EM-style algorithm that consists of learning variational dynamics and reward models that account for intrinsic uncertainty in the environment (E-step) and improve the objective w.r.t. a policy (M-step). Our comprehensive approach permits learning of risk-seeking and risk-averse policies for which the latter has been mostly ignored in the RL-as-inference literature. Our formulation also adapts to risk induced by stochastic rewards, a further extension of the RL-as-inference literature which assumes deterministic rewards. Furthermore, we demonstrate the robustness of our method to other risk-aware algorithms in risk-sensitive variants of Mujoco tasks. Code is available at https://github.com/AlonsoGranados/rsVAC/.

## 2 PRELIMINARIES: RISK-SENSITIVE REINFORCEMENT LEARNING

The RL framework consists of a Markov decision process (MDP) defined by a tuple $(\mathcal{S}, \mathcal{A}, p, \mathcal{R})$. $\mathcal{S}$, $\mathcal{A}$ and $\mathcal{R}$ are the state, action and reward spaces, respectively. The transition probability over the next state $s_{t+1} \in \mathcal{S}$ given the current state $s_t \in \mathcal{S}$ and action $a_t \in \mathcal{A}$ is denoted as $p(s_{t+1} \mid s_t, a_t)$, the initial state distribution as $p(s_1)$. A policy $\pi$ specifies a probability distribution over actions given a current state $s_t$. The reward $r_t \in \mathcal{R}$ is treated as a random variable with distribution $p(r_t|s_t, a_t)$. The distribution over trajectory $\tau = (s_1, a_1, r_1, s_2, a_2, ..., s_T, a_T, r_T, s_{T+1})$ for a sampling policy $\pi$ is given by $p_\pi(\tau) = p(s_1) \prod_t p(s_{t+1} \mid s_t, a_t) p(r_t \mid s_t, a_t) \pi(a_t \mid s_t)$. The standard objective in RL is to find a policy that maximizes expected return: $\pi^* = \arg\max_\pi \mathbb{E}_{p_\pi(\tau)}[\sum_{t=1}^T r_t]$.

### 2.1 ENTROPIC RISK MEASURE

In risk-sensitive RL with the entropic risk measure the goal is to find a policy that maximizes:

$$\max_\pi \beta \log \mathbb{E}_{p_\pi(\tau)} \left[ \exp\left( \frac{\sum_t r_t}{\beta} \right) \right], \tag{1}$$

for risk parameter $\beta \in \mathbb{R}$. This objective is closely related to mean-variance RL (Mannor & Tsitsiklis, 2011) given that a Taylor expansion of Eq. (1) yields $\mathbb{E}_{p_\pi(\tau)}[\sum_t r_t] + \frac{1}{2\beta}\text{Var}_\pi(\sum_t r_t) + O(\frac{1}{\beta^2})$ (Mihatsch & Neuneier, 2002; García & Fernández, 2015). The parameter $\beta$ controls the risk-sensitivity of the objective producing *risk-seeking* policies for $\beta > 0$ and *risk-averse* policies for $\beta < 0$. Additionally, it reduces to the standard (*risk-neutral*) RL objective when $|\beta| \to \infty$. Based on this framework, we define the soft value functions as the cumulative rewards under the entropic risk:

$$V_\pi(s) = \log \mathbb{E}_{p_\pi} \left[ \exp\left( \frac{\sum_t r_t}{\beta} \right) \mid s_1 = s \right], \quad Q_\pi(s, a) = \log \mathbb{E}_{p_\pi} \left[ \exp\left( \frac{\sum_t r_t}{\beta} \right) \mid s_1 = s, a_1 = a \right]. \tag{2}$$

These functions are recursively associated via Bellman-style backup equations:

$$V_\pi(s_t) = \log \mathbb{E}_{\pi(\cdot|s_t)} [\exp(Q_\pi(s_t, a_t))], \quad Q_\pi(s_t, a_t) = \log \mathbb{E}_{p(\cdot|s_t, a_t)} \left[ \exp\left( \frac{r_t}{\beta} + V_\pi(s_{t+1}) \right) \right]. \tag{3}$$

These are known as *soft* value functions, due to the presence of operators $\log \mathbb{E}[\exp(\cdot)]$ that act as soft approximations to $\max(\cdot)$. Finally, we have the Bellman optimality equations,

$$V^*(s_t) = \max_{a_t} Q^*(s_t, a_t), \quad Q^*(s_t, a_t) = \log \mathbb{E}_{p(\cdot|s_t, a_t)} \left[ \exp\left( \frac{r_t}{\beta} + V^*(s_{t+1}) \right) \right]. \tag{4}$$

Although these value functions can be estimated using dynamic programming, they require knowledge of the transition dynamics and the reward model to compute the expectations, since unbiased sample-based estimates are not available due to the nonlinear log operation. Fig. 1 illustrates the impact of the risk-sensitivity parameter $\beta$ on the optimal policy in a simple three arms MDP, and its effects in modulating risk-seeking and risk-averse policies. We emphasize that a risk-neutral policy is recovered for large $|\beta|$ values, while small $|\beta|$ values produce risk-seeking/averse policies.

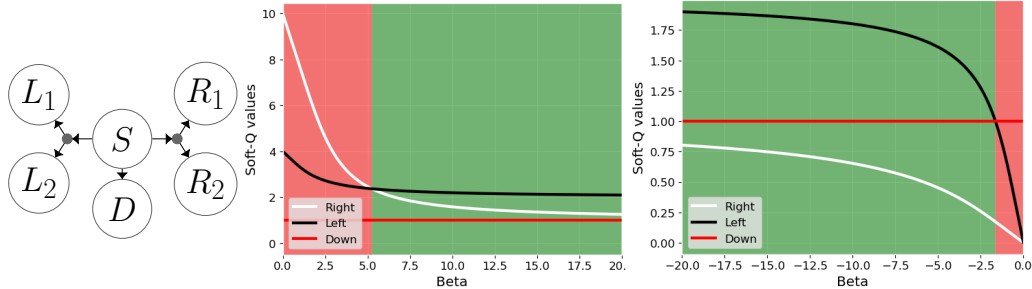

Figure 1: **Three arms environment** *Left:* MDP with three actions (left, down and right) and initial state S. Action 'right' produces reward 0 and 10 with probability 0.9 and 0.1, respectively. Action 'left' produces reward 0 and 4 with probability 0.5 and 0.5, respectively. Finally, action 'down' has a deterministic reward of 1. A risk neutral agent would prefer action 'left' which has the highest mean return while a risk-seeking and risk-averse agents would prefer action 'right' and 'down', respectively. *Middle:* Soft-Q values as a function of $\beta$ for $\beta > 0$. Observe that for small $\beta$ the framework learns a risky policy (red region) while for large $\beta$ it recovers an optimal risk neutral policy (green region). *Right:* Soft-Q values as a function of $\beta$ for $\beta < 0$. Now the agent learns a risk-averse policy when $|\beta|$ is small (red region) while it recovers the neutral policy for larger $|\beta|$ (green region).

## 2.2 RISK-SENSITIVE VARIATIONAL BOUND

In this work we leverage the well-established connection between RL and probabilistic inference (Levine, 2018). Under this formulation we incorporate the rewards into a probabilistic model by introducing a set of binary auxiliary variables $\mathcal{O}_t \in \{0, 1\}$ that are *independently* distributed at each time as $p(\mathcal{O}_t = 1 \mid r_t) \propto \exp(\frac{r_t}{\beta})$. The event $\mathcal{O}_t = 1$ can loosely be interpreted as the agent having acted optimally at time $t$[1]. An important motivation for using this interpretation is that we can define a surrogate objective for the entropic risk measure via the evidence lower bound (ELBO) on the log-marginal likelihood:

$$\log p_\pi(\mathcal{O}_{1:T}) = \log \mathbb{E}_{p_\pi}\left[\exp\left(\sum_t \frac{r_t}{\beta}\right)\right] \geq \mathbb{E}_q\left[\sum_t \frac{r_t}{\beta}\right] - \mathrm{KL}(q(\tau) \,\|\, p_\pi(\tau)) \coloneqq \mathcal{J}_\beta(q, \pi), \quad (5)$$

where the LHS is shorthand for the marginal likelihood of an optimal trajectory: $\log p_\pi(\mathcal{O}_{1:T} = 1)$. The log-marginal is equivalent to the entropic risk measure, up to a multiplicative constant $\beta$ which controls risk-sensitivity and is bounded by the RHS. The bound in Eq. (5) arises from the application of Jensen's inequality where $q(\tau)$ is a variational distribution over trajectories. This bound is tight when the variational distribution equals the posterior over trajectories $q(\tau) = p(\tau \mid \mathcal{O}_{1:T} = 1)$ almost everywhere. A variety of expectation-maximization (EM) style algorithms have been proposed to optimize $\mathcal{J}_\beta$ by alternating improvements w.r.t. $q$ and $\pi$ (Abdolmaleki et al., 2018b; Peters et al., 2010; Levine & Koltun, 2013; Chow et al., 2021).

## 3 VARIATIONAL MODEL / POLICY ITERATION

Using $\mathcal{J}_\beta(q, \pi)$ as a surrogate objective for the entropic risk measure we now propose two algorithms that approximately optimize it for, both, the risk-seeking ($\beta > 0$) and risk-averse ($\beta < 0$) settings. We consider variational distributions over trajectories of the form,

$$q_\pi(\tau) = p(s_1) \prod_{t=1}^T \pi(a_t|s_t) q_r(r_t|s_t, a_t) q_d(s_{t+1}|s_t, a_t). \quad (6)$$

Note that we incorporate a variational posterior distribution $q_r$ over rewards. This stochastic reward model is an extension of existing RL-as-inference methods that are constrained to deterministic rewards. Expanding the KL regularizer of Eq. (5) yields the variational objective,

$$\mathcal{J}_\beta(q, \pi) = \mathbb{E}_{q_\pi(\tau)}\left[\sum_t \frac{r_t}{\beta} - \log \frac{q_d(s_{t+1}|s_t, a_t)}{p(s_{t+1}|s_t, a_t)} - \log \frac{q_r(r_t|s_t, a_t)}{p(r_t|s_t, a_t)}\right]. \quad (7)$$

---

[1]The optimality interpretation is a loose one stemming from the reward at time $t$, which increases the probability of $\mathcal{O}_t = 1$ exponentially. This interpretation has become standard in the literature (Levine, 2018).

To optimize Eq. (7) we consider an EM-style algorithm where the E-step maximizes $\mathcal{J}_\beta$ w.r.t. $q$ and the M-step optimizes w.r.t. $\pi$. Risk-sensitivity arises in Eq. (7) from the maximization w.r.t. the variational distribution $q$. Although the penalty discourages deviations from the true model, the agent is willing to pay this penalty if the increase in expected return is large enough to compensate this extra cost. When $\beta > 0$, the variational model becomes optimistic (risk-seeking) as it aims to increase the expected return. When $\beta < 0$, it becomes pessimistic (risk-averse), as instead, it aims to increase the expected cost. Finally, we recover the true model when $|\beta| \to \infty$ as the objective only suffers the deviation penalty.

## 3.1 E-STEP

For a fixed policy $\pi$ we denote the optimal variational distribution as $q_\pi^* = \arg\max_q \mathcal{J}_\beta(q, \pi)$. Directly maximizing $\mathcal{J}_\beta$ can be computationally expensive as it requires optimizing $q$ over the full trajectory. Instead, we consider a Bellman-like operator $\mathcal{T}_\pi$ as a partial optimization over $q$ for a single transition where $V$ is a state-value function:

$$\mathcal{T}_\pi[V](s) = \mathbb{E}_{a \sim \pi(\cdot|s)}\left[\max_{q_r \in \Delta_\mathcal{R}} \mathbb{E}_{r \sim q_r}\left[\frac{r}{\beta} - \log\frac{q_r(r|s,a)}{p(r|s,a)}\right] + \max_{q_d \in \Delta_S} \mathbb{E}_{s' \sim q_d}\left[V(s') - \log\frac{q_d(s'|s,a)}{p(s'|s,a)}\right]\right]. \quad (8)$$

In particular, we have the following theorem for the operator $\mathcal{T}_\pi[V](s)$ (see Appendix for proof):

**Theorem 1.** *Repeated application of $\mathcal{T}_\pi$ to any value function $V$ such that $V(s_{T+1}) = 0$ converges to the optimal value function $V_k^*$ for all $k$, where:*

$$V_k^*(s_k) = \mathbb{E}_{q_\pi^*(\tau)}\left[\sum_{t=k}^T \frac{r_t}{\beta} - \log\frac{q_d^*(s_{t+1}|s_t, a_t)}{p(s_{t+1}|s_t, a_t)} - \log\frac{q_r^*(r_t|s_t, a_t)}{p(r_t|s_t, a_t)}\right]. \quad (9)$$

Hence, we can obtain the optimal value function by iteratively applying $\mathcal{T}_\pi$ to some initial value function $V_0$. We can recover the optimal variational distributions using the optimal value function:

**Theorem 2.** *Let $q_r^*$ and $q_d^*$ be the solution of $\arg\max_q \mathcal{J}_\beta(q, \pi)$. Then*

$$q_r^*(r|s,a) \propto p(r|s,a)\exp\left(\frac{r}{\beta}\right), \quad q_d^*(s'|s,a) \propto p(s'|s,a)\exp\left(V_\pi^*(s')\right). \quad (10)$$

All proofs can be found in the Appendix.

## 3.2 M-STEP

We optimize $\mathcal{J}(q^*, \pi)$ w.r.t. the policy $\pi$ using the variational distribution $q_\pi^*$ from the E-step:

$$\pi^* = \arg\max_\pi \mathbb{E}_{q_\pi^*(\tau)}\left[\sum_t \underbrace{r_t - \beta\log\frac{q_d^*(s_{t+1}|s_t, a_t)}{p(s_{t+1}|s_t, a_t)} - \beta\log\frac{q_r^*(r_t|s_t, a_t)}{p(r_t|s_t, a_t)}}_{:=\hat{r}_t}\right]. \quad (11)$$

Observe that Eq. (11) is equivalent to learning the optimal policy for a standard RL problem with transition dynamics $q$ and augmented rewards $\hat{r}_t$, so any RL algorithm can be used for the M-step. Although the expectation can now be estimated using easily-obtained samples from $q$ we still have the problem of needing to evaluate the dynamics and reward model in the augmented reward, which might be unknown. In the following section we address this by providing an algorithm that can be optimized using off-policy data. Finally, we note that in the risk-averse setting Eq. (11) corresponds to a minimization w.r.t. the policy: $\arg\min_\pi \mathcal{J}(q^*, \pi)$. See Appendix B for an extended discussion of optimization in the risk-averse regime.

## 4 RSVAC: RISK SENSITIVE VARIATIONAL ACTOR-CRITIC

We now present a practical RL algorithm that approximately optimizes $\mathcal{J}(q, \pi)$ using only collected experience by the agent. We make three design choices to approximate this objective: first, we learn

parameterized probabilistic networks, $p_\theta(s_{t+1}|s_t, a_t)$ and $p_\theta(r_t|s_t, a_t)$, for the unknown transition dynamics and reward model; next, we represent the variational distributions using probabilistic networks, $q_\phi(s_{t+1}|s_t, a_t)$ and $q_\phi(r_t|s_t, a_t)$, and approximate the maximization operation in the E-step with stochastic gradient descent; finally, we use an actor-critic architecture with function approximators to learn the optimal value function and policy from the M-step.

## 4.1 VARIATIONAL REWARD AND DYNAMICS MODEL OPTIMIZATION

We model the reward and dynamics as Gaussian distributions with mean and covariance given by neural networks and train them to minimize cross-entropy using stochastic gradient descent:

$$J_r(\theta) = -\mathbb{E}_{(s_t,a_t,r_t)\sim\mathcal{D}_{\text{env}}}\left[\log p_\theta(r_t|s_t,a_t)\right], J_d(\theta) = -\mathbb{E}_{(s_t,a_t,s_{t+1})\sim\mathcal{D}_{\text{env}}}\left[\log p_\theta(s_{t+1}|s_t,a_t)\right] \quad (12)$$

where $\mathcal{D}_{\text{env}}$ is an experience replay buffer that stores previously seen interactions with the environment. We similarly parameterize the variational models using Gaussian distributions. To learn the variational reward model we approximate the optimization w.r.t. $q_\phi(r_t|s_t, a_t)$ in Eq. (8) by maximizing:

$$J_r(\phi) = \mathbb{E}_{(s_t,a_t)\sim\mathcal{D}_{\text{env}},r_t\sim q_\phi(r_t|s_t,a_t)}\left[\frac{r_t}{\beta} - \log\frac{q_\phi(r_t|s_t,a_t)}{p_\theta(r_t|s_t,a_t)}\right], \quad (13)$$

w.r.t. its parameters $\phi$. In particular, we use the reparameterization trick to obtain a lower variance estimator that can be optimized using stochastic gradient ascent,

$$J_r(\phi) = \mathbb{E}_{(s_t,a_t)\sim\mathcal{D}_{\text{env}},\epsilon\sim\mathcal{N}}\left[\frac{f_\phi(\epsilon;s_t,a_t)}{\beta} - \log\frac{q_\phi(f_\phi(\epsilon;s_t,a_t)|s_t,a_t)}{p_\theta(f_\phi(\epsilon;s_t,a_t)|s_t,a_t)}\right], \quad (14)$$

where $f_\phi(\epsilon; s_t, a_t)$ is the reparameterized reward model and $\epsilon$ is a noise vector sampled from a spherical Gaussian distribution. Similarly, we learn variational dynamics by approximating the optimization w.r.t. $q_\phi(s_{t+1}|s_t, a_t)$ in Eq. (8) with:

$$J_d(\phi) = \mathbb{E}_{(s_t,a_t)\sim\mathcal{D}_{\text{env}},\epsilon\sim\mathcal{N}}\left[V_\psi(g_\phi(\epsilon;s_t,a_t)) + \log\frac{q_\phi(g_\phi(\epsilon;s_t,a_t)|s_t,a_t)}{p_\theta(g_\phi(\epsilon;s_t,a_t)|s_t,a_t)}\right], \quad (15)$$

where again we use the reparameterization trick to reparameterize the dynamics model, $s_{t+1} = g_\phi(\epsilon; s_t, a_t)$, and substitute the optimal value function with a critic $V_\psi$ that can be differentiated so Eq. (15) can be optimized using stochastic gradient ascent.

## 4.2 ACTOR-CRITIC OPTIMIZATION

We now present an actor-critic algorithm to optimize the M-step. As previously stated, the optimization in Eq. (11) is equivalent to the RL problem that has transition dynamics $q_\phi(s_{t+1}|s_t, a_t)$, reward model $q_\phi(r_t|s_t, a_t)$, and augmented reward $\hat{r}_t = r_t - \beta\log\frac{q_\phi(s_{t+1}|s_t,a_t)}{p_\theta(s_{t+1}|s_t,a_t)} - \beta\log\frac{q_\phi(r_t|s_t,a_t)}{p_\theta(r_t|s_t,a_t)}$. We collect transitions from the variational model using branched rollout (Janner et al., 2019), i.e. we sample states under the true dynamics $\mathcal{D}_{\text{env}}$ and run the policy under $q_\phi$ to generate new transitions which we store in the model replay buffer $\mathcal{D}_{\text{model}}$. We approximate the critic using a neural network $Q_\psi(s_t, a_t)$ which we train by minimizing the squared TD-error:

$$J_Q(\psi) = \mathbb{E}_{(s_t,a_t,r_t,s_{t+1})\sim\mathcal{D}_{\text{model}}}\left[\left(Q_\psi(s_t,a_t) - \hat{r}_t - V'_\psi(s_{t+1})\right)^2\right], \quad (16)$$

using stochastic gradient descent and samples from the model replay buffer. The optimal state-value function $V'_\psi(s_{t+1})$ is implicitly represented by $\mathbb{E}_{a_{t+1}\sim\pi_\theta(a_{t+1}|s_{t+1})}[Q'_\psi(s_{t+1}, a_{t+1})]$ where $Q'_\psi$ is a target critic network that we update using an exponentially moving average of the $Q_\psi$ weights (Lillicrap et al., 2015). We approximate this expectation using a single action sample from the policy $\pi_\theta$. The policy $\pi_\theta$ is a Gaussian distribution parameterized with neural networks and is trained to maximize $Q_\psi$ with an added entropy regularizer to improve exploration during learning (Haarnoja et al., 2018):

$$J_\pi(\theta) = \mathbb{E}_{s_t\sim\mathcal{D}_{\text{env}},\epsilon\sim\mathcal{N}}\left[Q_\psi(s_t,f_\theta(\epsilon;s_t)) - \log\pi_\theta(f_\theta(\epsilon;s_t)|s_t)\right] \quad (17)$$

where we have reparameterized the policy $a_t = f_\theta(\epsilon; s_t)$ and $\epsilon$ is a noise vector sampled from a spherical Gaussian distribution. Again we learn these parameters using stochastic gradient descent. One benefit of rsVAC is that it enjoys great flexibility so any actor-critic method can be incorporated into framework as long as the rewards and samples come from the variational model. Pseudocode for the rsVAC algorithm can be found in Appendix F.

## 5 RELATED WORK

**Entropic risk.** The risk-sensitive objective with entropic risk measure was first described by the seminal work of (Howard & Matheson, 1972). This work has inspired many methods in a variety of settings (Borkar, 2001; 2002; 2010; Borkar & Meyn, 2002; Coraluppi & Marcus, 1999; Di Masi & Stettner, 1999; Fleming & McEneaney, 1995; Hernández-Hernández & Marcus, 1996; Huang & Haskell, 2020). However, these algorithms are constrained to simple environments as they require knowledge of the transition dynamics or assume access to a simulator of the environment. In the setting with unknown transition dynamics, TD(0) and Q-learning-style algorithms have been proposed by applying an exponential transformation to the risk sensitive objective, but estimating these value functions can lead to instabilities when introducing function approximators (Bäuerle & Rieder, 2014; Borkar, 2002; Fei et al., 2021b;a; Mihatsch & Neuneier, 2002; Noorani et al., 2023).

**RL-as-inference.** Probabilistic inference methods for solving RL can be traced back to the Kalman-duality in linear-quadratic systems (Kalman, 1960) and later to linearly solvable MDPs (Todorov, 2006). The variational framework can be formulated as searching for maximum likelihood policies on an augmented MDP with exponentiated rewards treated as probabilities and is equivalent to the risk-sensitive objective (Todorov, 2008; Levine & Koltun, 2013; Levine, 2018). These approaches learn risk-seeking policies as they only consider $\beta > 0$. Model-free variational approaches such as MaxEnt RL combat this behavior by removing the controller's ability to modify the variational dynamics, resulting in high-entropy policies. This penalization of determinism has been effective in some high-dimensional tasks (O'Donoghue et al., 2016; Nachum et al., 2017; Haarnoja et al., 2018; Lee et al., 2020), but has been shown to lead to undesirable behavior (Fellows et al., 2019). Closely related KL-regularized RL methods include a proximal operator on the policy (Peters et al., 2010; Schulman et al., 2015; Chebotar et al., 2017; Noorani & Baras, 2021). More generally, EM-style algorithms jointly optimize their variational and prior policies (Peters & Schaal, 2007; Neumann et al., 2011; Abdolmaleki et al., 2018b;a). Our variational formulation is most similar to the one used in VMBPO (Chow et al., 2021), an EM-style algorithm that also learns variational dynamics. However, their approach only considers risk-seeking policies and deterministic rewards.

**Connections to other methods.** The role of the risk parameter $\beta$ is to limit the disagreement between the variational and true environment dynamics and reward model through the KL penalty. $\beta$-VAE (Higgins et al., 2016) studies a similar objective to ours, albeit in the different context of representation learning, where $\beta$ limits the capacity of the variational distribution to learn disentangled representations. In Bayesian RL, a similar risk parameter balances the exploration-exploitation trade-off by modulating *epistemic uncertainty* (O'Donoghue et al., 2019; O'Donoghue & Lattimore, 2021; O'Donoghue, 2023).

## 6 EXPERIMENTS

We evaluate the ability of rsVAC to learn risk-sensitive policies in a variety of risky environments. First, we consider a risky variation of the tabular environment discussed in Eysenbach et al. (2022), where exact inference is possible and equality to the entropic risk can be achieved. We next evaluate the inclusion of function approximators in a continuous 2D environment with stochastic transition dynamics where the goal is to land proximal to the environment boundary without crossing over. Finally, we compare rsVAC to risk-sensitive baseline methods in variations of three challenging MuJoCo environments that incorporate risk in the manner introduced by Luo et al. (2024). In all cases we find that rsVAC capably learns risk-sensitive policies in, both, the risk-averse and risk-seeking regimes while simultaneously achieving high reward.

### 6.1 RISK IN TABULAR ENVIRONMENTS

Our motivation for using the tabular setting is that we can study the risk-sensitive behavior of our algorithm without introducing function approximation error. We consider a risky variant of the gridworld presented in Eysenbach et al. (2022). In this environment the agent's goal, which starts from the top left corner, is to reach the star goal state (See Fig. 2a). We modify the original environment to incorporate aleatoric risk by including a cliff region (gray squares in Fig. 2a). Falling into the cliff incurs a large negative reward and transition to the initial state. The agent can choose

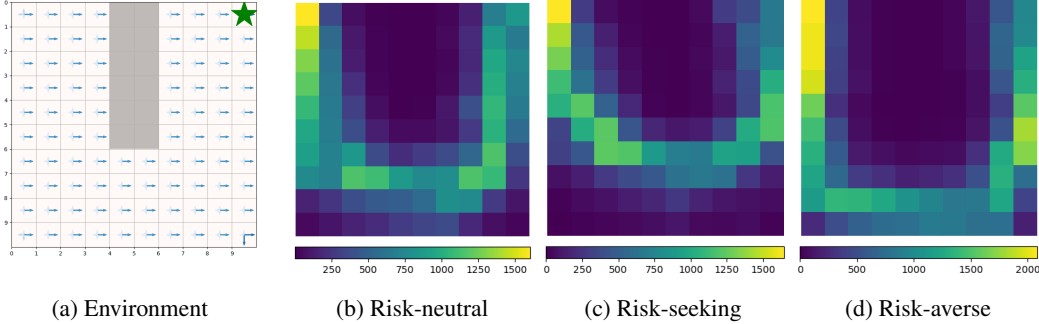

| (a) Environment | (b) Risk-neutral | (c) Risk-seeking | (d) Risk-averse |

Figure 2: **Risky tabular setting.** (a) The modified grid environment with cliff region given by gray states. We show the dynamics in the grid for the action 'right' at each state, where we represent a transition probability between two states as a vectors with its magnitude proportional to the probability. (b, c, d) To demonstrate the risk preferences of our algorithm, we sample 1000 episodes for three policies — Q-learning (risk-neutral), $\beta = 1$ (risk-seeking) and $\beta = -0.5$ (risk-averse) — and compute histograms for the count of visited states.

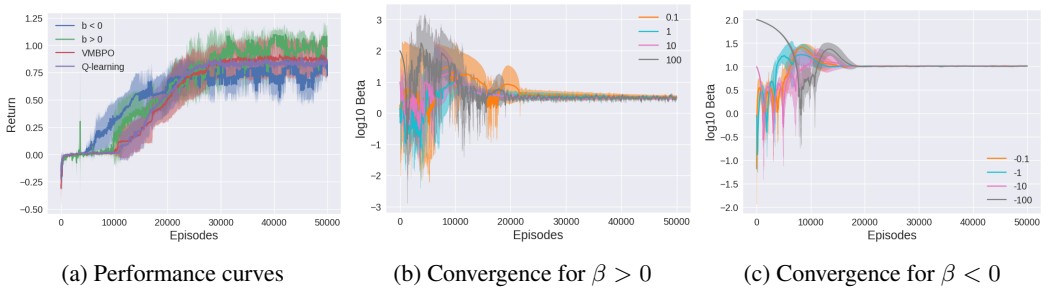

| (a) Performance curves | (b) Convergence for $\beta > 0$ | (c) Convergence for $\beta < 0$ |

Figure 3: **Stochastic cliff performance.** (a): We compare the expected return for 5 independent runs for different algorithms. rsVAC for $\beta > 0$ performs comparably to both Q-learning and VMBPO. (b,c): We show that dual optimization eventually converges to the same optimal value for different initial settings of $\beta$.

from four actions (up, left, down and right) which can result in a transition to the chosen direction or moving randomly to one of the four directions with equal probability.

We demonstrate that rsVAC can produce risk-sensitive policies by training the surrogate objective for the values of $\beta = 1$ and $\beta = -0.5$, along with the risk-neutral policy. We compare the risk preferences of these policies by computing a histogram over states for 1000 trajectories. From these trajectories, we observe that the risk-seeking policy (Fig. 2c) takes the shortest path to the goal, but in the process occasionally falls into the cliff. In contrast, the risk-averse policy (Fig. 2d) avoids entirely the cliff region resulting in longer trajectories. Finally, the risk-neutral policy takes a middle-of-the-road approach between the two previous policies (Fig. 2b) where it rarely falls into the cliff but on average it takes to longer to reach the final state in comparison to the risk-seeking policy.

We now compare the average return performance of rsVAC when including dual optimization w.r.t. $\beta$ discussed in Appendix C. As comparison baselines we consider Q-learning and VMBPO (Chow et al., 2021), a model-based algorithm that also learns variational dynamics but is restricted to the risk-seeking setting. The performance curves in Fig. 3a show that rsVAC ($\beta > 0$) is as efficient and performs as well or better than VMBPO and Q-learning. Figs. 3b and 3c demonstrate robustness of our dual optimization over $\beta$, which converges to the same value regardless of initialization.

## 6.2 STOCHASTIC CONTINUOUS 2D ENVIRONMENT

We verify that our algorithm can learn risk-sensitive policies with function approximators in a stochastic continuous environment. An agent begins in the middle of a 2D space and the goal is to navigate as near to the lower left-or-right corners as possible without crossing the side edges. The agent observes its 2D coordinates $(x, y)$, chooses a direction $a$ ($||a||_2 = 1$), and moves in that

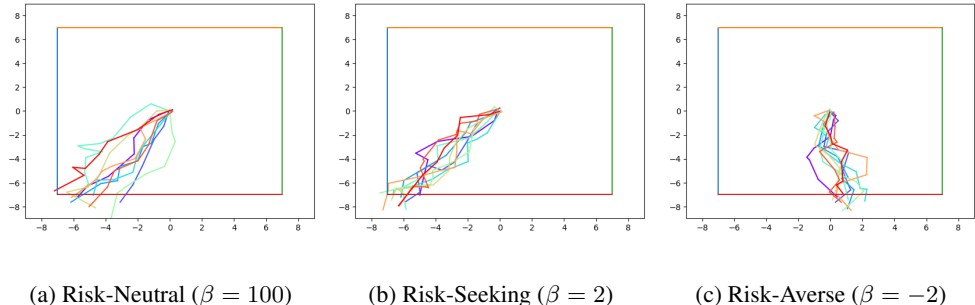

(a) Risk-Neutral ($\beta = 100$)          (b) Risk-Seeking ($\beta = 2$)          (c) Risk-Averse ($\beta = -2$)

Figure 4: **Trajectories for stochastic 2D environment.** We illustrate the learned policy by sampling 10 trajectories for different $\beta$ values. (a) Policies trained with large $\beta$ magnitude tend to be risk-neutral. (b) Policies trained with small positive $\beta$ are risk-seeking and try to hit high reward ignoring potentially hitting the side wall. (c) Policies trained with negative $\beta$ stay in the center part of the square.

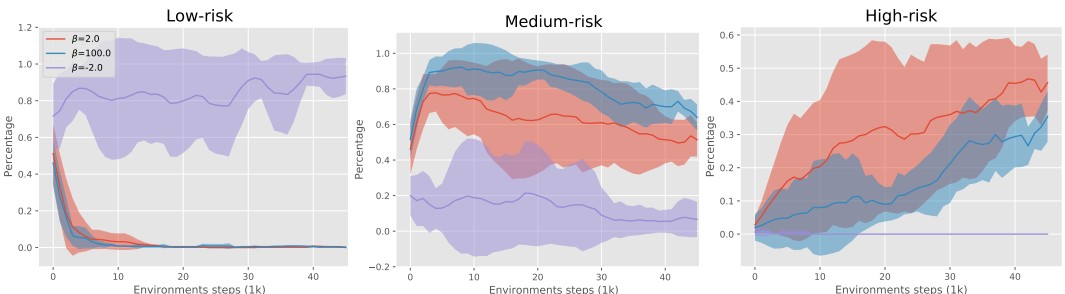

Figure 5: **Exit regions for stochastic 2D environment.** We define 3 regions depended on the agent's X-position when an episode ends: low-risk ($|x| < 2.8$), medium-risk ($2.8 \leq |x| < 5.6$), and high-risk ($5.6 \leq |x| < 7$). We calculate these percentage regions as a function of environment steps over different $\beta$ values.

direction with noise sampled from $\mathcal{N}(a, 0.5^2 I)$. An episode terminates when the agent leaves the square given by $\{(x,y) : |x| \leq 7, |y| \leq 7\}$. The agent receives $-0.1$ reward at every step with an additional positive reward proportional to its X-position ($(100/7) * |x|$) when it exits the square through bottom, or $-100$ reward if it leaves through either the left-or-right side of the square.

In Fig. 4, we visualize trajectories for the different learned policies on the true environment. Observe that small positive $\beta$ values tend to produce risk-seeking policies where the agent aims to get as much reward (close to the walls) as possible while ignoring the likelihood of hitting the sides of the square. Policies trained with negative $\beta$ produce risk-avoiding policies that stay in the center region. We also calculate the percentage of episodes that terminate in different risk-regions (Fig. 5) which demonstrate how $\beta$ interpolates between different regions. We designate low-risk (left) as far from the walls, medium-risk (center), and high-risk (right) as near the wall. We find that negative $\beta$ primarily terminate in the low-risk region, whereas small positive $\beta$ primarily terminates in the high-risk region and risk-neutral ($\beta = 100$) terminates in the intermediate and to a lesser extent high-risk regions. We additionally include the visualizations for the learned variational dynamics in the Appendix (Fig. 7).

### 6.3    SIMULATED ROBOTIC BENCHMARKS

We use the MuJoCo physics engine (Todorov et al., 2012) in Gymnasium (Towers et al., 2023) to evaluate our method on three continuous tasks (InvertedPendulum, HalfCheetah, and Swimmer). We follow the modifications made to the reward function as in (Luo et al., 2024) to produce risky regions in these environments based on the X-position of the agent. An additional stochastic reward is sampled from $\mathcal{N}(0, 10^2)$ if X-position $> 0.01$ in InvertedPendulum, $> -3$ in HalfCheetah, or $> 0.5$ in Swimmer. Hence, we can test whether an agent is risk-seeking or risk-averse by calculating the percentage of time spent in the region of stochastic reward.

We compare our method to **Mean Gini deviation (MG)** (Luo et al., 2024), a policy gradient algorithm that optimizes the Gini deviation as an alternative risk measure and outperforms other mean-variance

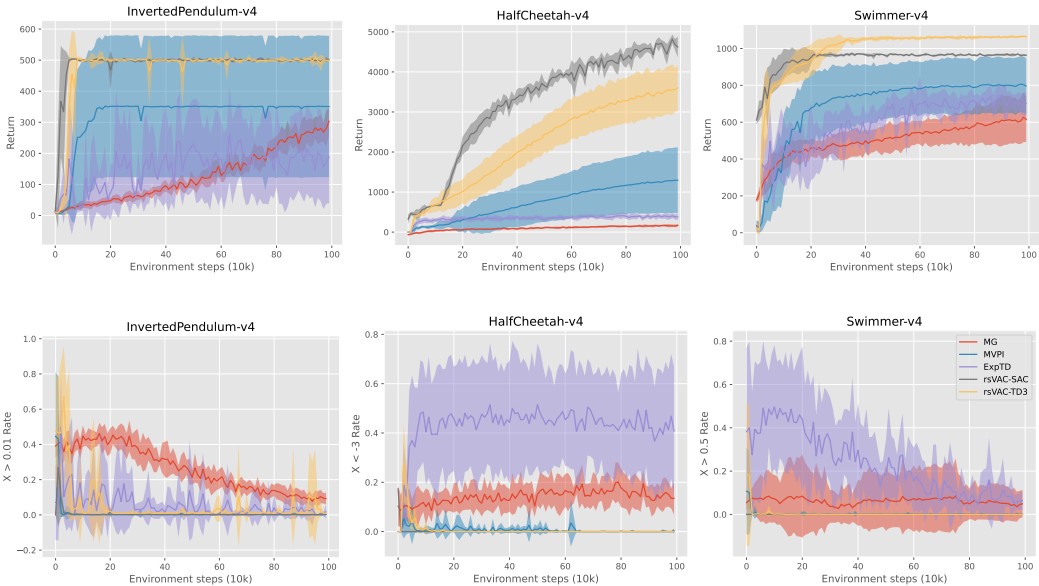

Figure 6: **Risk-Averse MuJoCo.** *Top row*: Average return on risky MuJoCo benchmarks. *Bottom row*: Percentage of steps on an episode in risky regions. The solid curves correspond to the mean and shaded regions to $\pm$ one standard deviation over 10 random trials.

algorithms; **mean-variance policy iteration (MVPI)** (Zhang et al., 2021), a highly flexible algorithm that optimizes reward-volatility risk measure with the primary goal of reducing the performance gap between risk-neutral and risk-averse algorithms; and **exponential TD (expTD)** (Noorani et al., 2023), an actor-critic algorithm that optimizes the entropic risk-measure by using a critic that estimates the exponentiated return. To achieve a fair comparison, we implement every actor-critic algorithm on top of TD3 (Fujimoto et al., 2018). For MG we follow the author implementation which uses PPO-style policy gradient to maximize the expected return (Schulman et al., 2017). For consistency we use the same network architectures across all algorithms. We also update the policy at each environment step for all algorithms, with the exception of MG which requires the collection of 10 episodes before updating its model. See the Appendix for additional configuration details.

In Fig. 6, we report the total return (*top-row*) and percentage of timesteps visiting the noisy region (*bottom-row*) for each algorithm in a risk-averse configuration. We perform 10 runs of each algorithm with different random seeds and report average and STDEV every 10k environment steps. Agents are tested for 20 episodes per evaluation. For rsVAC, we include both a version where the actor-critic is given by TD3 and another given by SAC (Haarnoja et al., 2018). The results show that rsVAC is effective at learning the stochasticity of the environment while producing better policies in terms of learning speed and final-performance. MVPI also learns risk-averse policies in all three domains, but results in lower overall mean returns. We also perform an ablation analysis for the parameter $\beta$ to demonstrate that our algorithm can learn both risk-seeking and risk-averse policies (Appendix Fig. 8).

## 7 CONCLUSION

In this work, we leveraged the connection between RL and probabilistic inference to formulate a surrogate objective on the entropic risk measure. We proposed an EM-style algorithm that consists of learning variational dynamics and reward model that account for aleatoric uncertainty in the environment (E-step) and improves the objective w.r.t. a policy (M-step). Finally, we proposed a practical algorithm (rsVAC) that permits learning of risk-seeking and risk-averse policies from experience replay alone. Our evaluations demonstrate that rsVAC is effective in learning risk-sensitive policies in several challenging environments. In particular we show that, compared to baseline risk-sensitive methods, rsVAC performs capably in risky variants of challenging MuJoCo environments, and in all cases yields superior return.

## 8 ACKNOWLEDGMENTS

This material is based upon work supported by the Air Force Office of Scientific Research under award number FA9550-22-1-0194.

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

APPENDIX

## A  OPERATOR $\mathcal{T}_\pi$ PROOFS

For our formulation, we consider a finite-horizon problem for which we maximize the following objective over variational distributions $q_d^t(s_{t+1}|s_t, a_t)$ and $q_r^t(r_t|s_t, a_t)$:

$$V_1^*(s_1) = \max_{q_r^1, q_d^1, \ldots, q_r^T, q_d^T} \mathbb{E}_{q_\pi(\tau)} \left[ \sum_{t=1}^T \frac{r_t}{\beta} - \log \frac{q_d^t(s_{t+1}|s_t, a_t)}{p_d^t(s_{t+1}|s_t, a_t)} - \log \frac{q_r^t(r_t|s_t, a_t)}{p_r^t(r_t|s_t, a_t)} \right]. \quad (18)$$

Let $\{q_r^{1*}, q_d^{1*}, \ldots, q_r^{T*}, q_d^{T*}\}$ be an optimal set of variational distributions for this problem. By the principle of optimality, we have that the truncated set of variational distributions $\{q_r^{k*}, q_d^{k*}, \ldots, q_r^{T*}, q_d^{T*}\}$ is optimal for the subproblem where we start at $s_k$:

$$V_k^*(s_k) = \max_{q_r^k, q_d^k, \ldots, q_r^T, q_d^T} \mathbb{E}_{q_\pi(\tau)} \left[ \sum_{t=k}^T \frac{r_t}{\beta} - \log \frac{q_d^t(s_{t+1}|s_t, a_t)}{p_d^t(s_{t+1}|s_t, a_t)} - \log \frac{q_r^t(r_t|s_t, a_t)}{p_r^t(r_t|s_t, a_t)} \right]. \quad (19)$$

**Lemma 1.** *Define the recursive value functions as:*

$$V_T(s_T) = \mathbb{E}_{a_T \sim \pi} \left[ \max_{q_r^T} \mathbb{E}_{q_r^T} \left[ \frac{r_T}{\beta} - \log \frac{q_r^T(r_T|s_T, a_T)}{p_r^T(r_T|s_T, a_T)} \right] + \max_{q_d^T} \mathbb{E}_{q_d^T} \left[ -\log \frac{q_d^T(s_{T+1}|s_T, a_T)}{p_d^T(s_{T+1}|s_T, a_T)} \right] \right].$$

$$V_k(s_k) = \mathbb{E}_{a_k \sim \pi} \left[ \max_{q_r^k} \mathbb{E}_{q_r^k} \left[ \frac{r_k}{\beta} - \log \frac{q_r^k(r_k|s_k, a_k)}{p_r^k(r_k|s_k, a_k)} \right] + \max_{q_d^k} \mathbb{E}_{q_d^k} \left[ V_{k+1}(s_{k+1}) - \log \frac{q_d^k(s_{k+1}|s_k, a_T)}{p_d^k(s_{k+1}|s_k, a_k)} \right] \right]$$

$$k = 1, \ldots, T - 1.$$

*Then we have that $V_k(s_k) = V_k^*(s_k)$*

*Proof.* This analysis proceeds similar to the proof of Prop. 1.3.1 of Bertsekas (2020). We will show by induction that the functions $V_k$ are equal to the optimal value functions $V_k^*$. For $k = T$, we have that

$$V_T^*(s_T) = \max_{q_r^T, q_d^T} \mathbb{E}_{a_T \sim \pi} \left[ \mathbb{E}_{q_r^T} \left[ \frac{r_T}{\beta} - \log \frac{q_r^T(r_T|s_T, a_T)}{p_r^T(r_T|s_T, a_T)} \right] + \mathbb{E}_{q_d^T} \left[ -\log \frac{q_d^T(s_{T+1}|s_T, a_T)}{p_d^T(s_{T+1}|s_T, a_T)} \right] \right]$$

$$= \mathbb{E}_{a_T \sim \pi} \left[ \max_{q_r^T} \mathbb{E}_{q_r^T} \left[ \frac{r_T}{\beta} - \log \frac{q_r^T(r_T|s_T, a_T)}{p_r^T(r_T|s_T, a_T)} \right] + \max_{q_d^T} \mathbb{E}_{q_d^T} \left[ -\log \frac{q_d^T(s_{T+1}|s_T, a_T)}{p_d^T(s_{T+1}|s_T, a_T)} \right] \right] = V_T(s_T),$$

where the max and expectation operators commute by the principle of optimality. Now, let us assume that for some $k$ and all $s_{k+1}$, we have $V_{k+1}(s_{k+1}) = V_{k+1}^*(s_{k+1})$. Then

$$V_k^*(s_k) = \max_{q_r^k, q_d^k, \ldots, q_r^T, q_d^T} \mathbb{E}_{q_\pi(\tau)} \left[ \sum_{t=k}^T \frac{r_t}{\beta} - \log \frac{q_d^t(s_{t+1}|s_t, a_t)}{p_d^t(s_{t+1}|s_t, a_t)} - \log \frac{q_r^t(r_t|s_t, a_t)}{p_r^t(r_t|s_t, a_t)} \right]$$

$$= \mathbb{E}_{a_k \sim \pi} \left[ \max_{q_r^k} \mathbb{E}_{q_r^k} \left[ \frac{r_k}{\beta} - \log \frac{q_r^k(r_k|s_k, a_k)}{p_r^k(r_k|s_k, a_k)} \right] \right.$$

$$+ \max_{q_d^k} \mathbb{E}_{q_d^k} \left[ \max_{q_r^{k+1}, \ldots, q_d^T} \mathbb{E}_{q_\pi} \left[ \sum_{t=k+1}^T \frac{r_t}{\beta} - \log \frac{q_d^t(s_{t+1}|s_t, a_t)}{p_d^t(s_{t+1}|s_t, a_t)} - \log \frac{q_r^t(r_t|s_t, a_t)}{p_r^t(r_t|s_t, a_t)} \right] - \log \frac{q_d^k(s_{k+1}|s_k, a_k)}{p_d^k(s_{k+1}|s_k, a_k)} \right] \right]$$

$$= \mathbb{E}_{a_k \sim \pi} \left[ \max_{q_r^k} \mathbb{E}_{q_r^k} \left[ \frac{r_k}{\beta} - \log \frac{q_r^k(r_k|s_k, a_k)}{p_r^k(r_k|s_k, a_k)} \right] + \max_{q_d^k} \mathbb{E}_{q_d^k} \left[ V_{k+1}^*(s_{k+1}) - \log \frac{q_d^k(s_{k+1}|s_k, a_k)}{p_d^k(s_{k+1}|s_k, a_k)} \right] \right]$$

$$= \mathbb{E}_{a_k \sim \pi} \left[ \max_{q_r^k} \mathbb{E}_{q_r^k} \left[ \frac{r_k}{\beta} - \log \frac{q_r^k(r_k|s_k, a_k)}{p_r^k(r_k|s_k, a_k)} \right] + \max_{q_d^k} \mathbb{E}_{q_d^k} \left[ V_{k+1}(s_{k+1}) - \log \frac{q_d^k(s_{k+1}|s_k, a_k)}{p_d^k(s_{k+1}|s_k, a_k)} \right] \right] = V_k(s_k).$$

where we obtain the second equality by moving the max operator inside the expectation using the principle of optimality. For the third equality, we use the definition of $V_{k+1}^*$, and for the fourth equality we use the induction hypothesis. This completes our induction, and we have $V_k^*(s_k) = V_k(s_k)$ for all $k$. $\qquad\square$

The ability to commute expectation and maximization operators, used in the previous proof, deserves additional discussion. Formal proofs of this result in the general DP setting can be found in appendix A of Bertsekas (2012). We include a discussion specific to our setting below, beginning with the assumption that each maximization step is finite:

$$
Q_k(s_k, a_k) = \max_{q_r^k, q_d^k, \dots, q_r^T, q_d^T} \mathbb{E}_{q_\pi(\tau)} \left[ \sum_{t=k}^{T} \frac{r_t}{\beta} - \log \frac{q_d^t(s_{t+1}|s_t, a_t)}{p_d^t(s_{t+1}|s_t, a_t)} - \log \frac{q_r^t(r_t|s_t, a_t)}{p_r^t(r_t|s_t, a_t)} \right] < \infty.
$$

Hence, for every $\epsilon > 0$ there exist a set of variational distributions $\{q_r^{k\epsilon}, \dots, q_d^{T\epsilon}\}$ that satisfies that

$$
\mathbb{E}_{q_\pi^\epsilon(\tau)} \left[ \sum_{t=k}^{T} \frac{r_t}{\beta} - \log \frac{q_d^{t\epsilon}(s_{t+1}|s_t, a_t)}{p_d^t(s_{t+1}|s_t, a_t)} - \log \frac{q_r^{t\epsilon}(r_t|s_t, a_t)}{p_r^t(r_t|s_t, a_t)} \right] \geq Q_k(s_k, a_k) - \epsilon.
$$

Then we have that,

$$
\max_{q_r^k, q_d^k, \dots, q_r^T, q_d^T} \mathbb{E}_{a_k \sim \pi} \left[ \mathbb{E}_{q_\pi(\tau)} \left[ \sum_{t=k}^{T} \frac{r_t}{\beta} - \log \frac{q_d^t(s_{t+1}|s_t, a_t)}{p_d^t(s_{t+1}|s_t, a_t)} - \log \frac{q_r^t(r_t|s_t, a_t)}{p_r^t(r_t|s_t, a_t)} \right] \right]
$$

$$
\geq \mathbb{E}_{a_k \sim \pi} \left[ \mathbb{E}_{q_\pi^\epsilon(\tau)} \left[ \sum_{t=k}^{T} \frac{r_t}{\beta} - \log \frac{q_d^{t\epsilon}(s_{t+1}|s_t, a_t)}{p_d^t(s_{t+1}|s_t, a_t)} - \log \frac{q_r^{t\epsilon}(r_t|s_t, a_t)}{p_r^t(r_t|s_t, a_t)} \right] \right]
$$

$$
\geq \mathbb{E}_{a_k \sim \pi} \left[ \max_{q_r^k, q_d^k, \dots, q_r^T, q_d^T} \mathbb{E}_{q_\pi(\tau)} \left[ \sum_{t=k}^{T} \frac{r_t}{\beta} - \log \frac{q_d^t(s_{t+1}|s_t, a_t)}{p_d^t(s_{t+1}|s_t, a_t)} - \log \frac{q_r^t(r_t|s_t, a_t)}{p_r^t(r_t|s_t, a_t)} \right] \right] - \epsilon.
$$

Since $\epsilon > 0$ is arbitrary it follows that

$$
\max_{q_r^k, q_d^k, \dots, q_r^T, q_d^T} \mathbb{E}_{a_k \sim \pi} \left[ \mathbb{E}_{q_\pi(\tau)} \left[ \sum_{t=k}^{T} \frac{r_t}{\beta} - \log \frac{q_d^t(s_{t+1}|s_t, a_t)}{p_d^t(s_{t+1}|s_t, a_t)} - \log \frac{q_r^t(r_t|s_t, a_t)}{p_r^t(r_t|s_t, a_t)} \right] \right]
$$

$$
\geq \mathbb{E}_{a_k \sim \pi} \left[ \max_{q_r^k, q_d^k, \dots, q_r^T, q_d^T} \mathbb{E}_{q_\pi(\tau)} \left[ \sum_{t=k}^{T} \frac{r_t}{\beta} - \log \frac{q_d^t(s_{t+1}|s_t, a_t)}{p_d^t(s_{t+1}|s_t, a_t)} - \log \frac{q_r^t(r_t|s_t, a_t)}{p_r^t(r_t|s_t, a_t)} \right] \right].
$$

On the other hand, we have that

$$
\mathbb{E}_{a_k \sim \pi} \left[ \max_{q_r^k, q_d^k, \dots, q_r^T, q_d^T} \mathbb{E}_{q_\pi(\tau)} \left[ \sum_{t=k}^{T} \frac{r_t}{\beta} - \log \frac{q_d^t(s_{t+1}|s_t, a_t)}{p_d^t(s_{t+1}|s_t, a_t)} - \log \frac{q_r^t(r_t|s_t, a_t)}{p_r^t(r_t|s_t, a_t)} \right] \right]
$$

$$
\geq \mathbb{E}_{a_k \sim \pi} \left[ \mathbb{E}_{q_\pi(\tau)} \left[ \sum_{t=k}^{T} \frac{r_t}{\beta} - \log \frac{q_d^t(s_{t+1}|s_t, a_t)}{p_d^t(s_{t+1}|s_t, a_t)} - \log \frac{q_r^t(r_t|s_t, a_t)}{p_r^t(r_t|s_t, a_t)} \right] \right]
$$

for all sets of variational distributions $\{q_r^k, \dots, q_d^T\}$. So the inequality holds when taking the maximum,

$$
\mathbb{E}_{a_k \sim \pi} \left[ \max_{q_r^k, q_d^k, \dots, q_r^T, q_d^T} \mathbb{E}_{q_\pi(\tau)} \left[ \sum_{t=k}^{T} \frac{r_t}{\beta} - \log \frac{q_d^t(s_{t+1}|s_t, a_t)}{p_d^t(s_{t+1}|s_t, a_t)} - \log \frac{q_r^t(r_t|s_t, a_t)}{p_r^t(r_t|s_t, a_t)} \right] \right]
$$

$$
\geq \max_{q_r^k, q_d^k, \dots, q_r^T, q_d^T} \mathbb{E}_{a_k \sim \pi} \left[ \mathbb{E}_{q_\pi(\tau)} \left[ \sum_{t=k}^{T} \frac{r_t}{\beta} - \log \frac{q_d^t(s_{t+1}|s_t, a_t)}{p_d^t(s_{t+1}|s_t, a_t)} - \log \frac{q_r^t(r_t|s_t, a_t)}{p_r^t(r_t|s_t, a_t)} \right] \right].
$$

Combining these two results we obtain the sought equality:

$$
\mathbb{E}_{a_k \sim \pi} \left[ \max_{q_r^k, q_d^k, \dots, q_r^T, q_d^T} \mathbb{E}_{q_\pi(\tau)} \left[ \sum_{t=k}^{T} \frac{r_t}{\beta} - \log \frac{q_d^t(s_{t+1}|s_t, a_t)}{p_d^t(s_{t+1}|s_t, a_t)} - \log \frac{q_r^t(r_t|s_t, a_t)}{p_r^t(r_t|s_t, a_t)} \right] \right]
$$

$$
= \max_{q_r^k, q_d^k, \dots, q_r^T, q_d^T} \mathbb{E}_{a_k \sim \pi} \left[ \mathbb{E}_{q_\pi(\tau)} \left[ \sum_{t=k}^{T} \frac{r_t}{\beta} - \log \frac{q_d^t(s_{t+1}|s_t, a_t)}{p_d^t(s_{t+1}|s_t, a_t)} - \log \frac{q_r^t(r_t|s_t, a_t)}{p_r^t(r_t|s_t, a_t)} \right] \right].
$$

We now present our main result for application of the Bellman-style operator.

**Theorem 1.** *Repeated application of $\mathcal{T}_\pi$ to any value function $V$ such that $V(s_{T+1}) = 0$ converges to the optimal value function $V_k^*$ for all $k$.*

*Proof.* We demonstrate this by induction. After one application of $\mathcal{T}_\pi$ we have that

$$\mathcal{T}_\pi[V](s_T) = \mathbb{E}_\pi \left[ \max_{q_r^T} \mathbb{E}_{q_r^T} \left[ \frac{r_T}{\beta} - \log \frac{q_r^T(r_T|s_T, a_T)}{p_r^T(r_T|s_T, a_T)} \right] + \max_{q_d^T} \mathbb{E}_{q_d^T} \left[ V(s_{T+1}) - \log \frac{q_d^T(s_{T+1}|s_T, a_T)}{p_d^T(s_{T+1}|s_T, a_T)} \right] \right].$$

$$= \mathbb{E}_\pi \left[ \max_{q_r^T} \mathbb{E}_{q_r^T} \left[ \frac{r_T}{\beta} - \log \frac{q_r^T(r_T|s_T, a_T)}{p_r^T(r_T|s_T, a_T)} \right] + \max_{q_d^T} \mathbb{E}_{q_d^T} \left[ - \log \frac{q_d^T(s_{T+1}|s_T, a_T)}{p_d^T(s_{T+1}|s_T, a_T)} \right] \right] = V_T(s_T),$$

where the second equality uses the fact that $V(s_{T+1}) = 0$. Now, let us assume that for some $k$ and all $s_{k+1}$, we have $V(s_{k+1}) = V_{k+1}(s_{k+1})$. Then

$$\mathcal{T}_\pi[V](s_k) = \mathbb{E}_\pi \left[ \max_{q_r^k} \mathbb{E}_{q_r^k} \left[ \frac{r_k}{\beta} - \log \frac{q_r^k(r_k|s_k, a_k)}{p_r^k(r_k|s_k, a_k)} \right] + \max_{q_d^k} \mathbb{E}_{q_d^k} \left[ V(s_{k+1}) - \log \frac{q_d^k(s_{k+1}|s_k, a_k)}{p_d^k(s_{k+1}|s_k, a_k)} \right] \right].$$

$$= \mathbb{E}_\pi \left[ \max_{q_r^k} \mathbb{E}_{q_r^k} \left[ \frac{r_k}{\beta} - \log \frac{q_r^k(r_k|s_k, a_k)}{p_r^k(r_k|s_k, a_k)} \right] + \max_{q_d^k} \mathbb{E}_{q_d^k} \left[ V_{k+1}(s_{k+1}) - \log \frac{q_d^k(s_{k+1}|s_k, a_k)}{p_d^k(s_{k+1}|s_k, a_k)} \right] \right] = V_k(s_k),$$

where the second equality uses the induction hypothesis. This shows that after $k$ successive applications of $\mathcal{T}_\pi$ we recover the optimal value function $V_k$. □

**Lemma 2.** *For any state $s \in \mathcal{S}$, action $a \in \mathcal{A}$ and reward distribution $p(r|s, a)$, we have*

$$\max_{q_r \in \Delta_\mathcal{R}} \mathbb{E}_{r \sim q_r(r|s,a)} \left[ \frac{r}{\beta} - \log \frac{q_r(r|s, a)}{p(r|s, a)} \right] = \log \mathbb{E}_{r \sim p(r|s,a)} \left[ \exp \left( \frac{r}{\beta} \right) \right]. \tag{20}$$

*Analogously, for any state $s \in \mathcal{S}$, action $a \in \mathcal{A}$, value function $V$ and dynamics distribution $p(s'|s, a)$, we have*

$$\max_{q_d \in \Delta_\mathcal{S}} \mathbb{E}_{s' \sim q_d(s'|s,a)} \left[ V(s') - \log \frac{q_d(s'|s, a)}{p(s'|s, a)} \right] = \log \mathbb{E}_{s' \sim p(s'|s,a)} [\exp(V(s'))]. \tag{21}$$

*Proof.* For Eq. 20, we have that

$$\max_{q_r \in \Delta_\mathcal{R}} \mathbb{E}_{r \sim q_r(r|s,a)} \left[ \frac{r}{\beta} - \log \frac{q_r(r|s, a)}{p(r|s, a)} \right] = \max_{q_r \in \Delta_\mathcal{R}} \mathbb{E}_{r \sim q_r(r|s,a)} \left[ \frac{r}{\beta} + \log p(r|s, a) - \log q_r(r|s, a) \right]$$

$$= \log \int_r \exp \left( \frac{r}{\beta} + \log p(r|s, a) \right) = \log \mathbb{E}_{r \sim p(r|s,a)} \left[ \exp \left( \frac{r}{\beta} \right) \right]$$

where the second equality follows from Lemma 4 in Nachum et al. (2017). Eq. 21 follows from Lemma 5 in Chow et al. (2021). □

**Lemma 3.** *The operator $\mathcal{T}_\pi$ is monotonic.*

*Proof.* If $V, W : \mathcal{S} \to \mathbb{R}$ are functions such that $V(s) \leq W(s), \forall s \in \mathcal{S}$. Then $\mathcal{T}_\pi[V](s) \leq \mathcal{T}_\pi[W](s), \forall s \in \mathcal{S}$. From Lemma 2, we have that

$$\mathcal{T}_\pi[V](s) = \mathbb{E}_\pi \left[ \log \mathbb{E}_{r \sim p} \left[ \exp \left( \frac{r}{\beta} \right) \right] + \log \mathbb{E}_{s' \sim p} [\exp(V(s'))] \right] = \mathbb{E}_\pi \left[ \log \mathbb{E}_{s', r \sim p} \left[ \exp \left( \frac{r}{\beta} + V(s') \right) \right] \right],$$

where the second equality uses the fact that $p(s', r|s, a) = p(s'|s, a)p(r|s, a)$. Therefore,

$$\mathcal{T}_\pi[V](s) = \mathbb{E}_\pi \left[ \log \mathbb{E}_{s', r \sim p} \left[ \exp \left( \frac{r}{\beta} + V(s') \right) \right] \right] \leq \mathbb{E}_\pi \left[ \log \mathbb{E}_{s', r \sim p} \left[ \exp \left( \frac{r}{\beta} + W(s') \right) \right] \right] = \mathcal{T}_\pi[W](s),$$

where we use the monotonicity of the $\exp$, expectation and $\log$ operations. □

Before proving Theorem 2, we prove the following lemma:

**Lemma 4.** *Let $q_r^*$ be the solution of Eq. 20. Then*

$$q_r^*(r|s,a) \propto p(r|s,a) \exp\left(\frac{r}{\beta}\right). \tag{22}$$

*Analogously, let $q_d^V$ be the solution of Eq. 21. Then*

$$q_d^V(s'|s,a) \propto p(s'|s,a) \exp\left(V(s')\right). \tag{23}$$

*Proof.* For Eq. 22, we have that

$$q_r^*(r|s,a) = \frac{\exp\left(\frac{r}{\beta} + \log p(r|s,a)\right)}{\int_r \exp\left(\frac{r}{\beta} + \log p(r|s,a)\right)} \propto p(r|s,a) \exp\left(\frac{r}{\beta}\right) \tag{24}$$

where the first equality follows from Corollary 6 in Nachum et al. (2017). Eq. 23 follows from Lemma 3 in Chow et al. (2021). □

**Theorem 2.** *Let $q_r^*$ and $q_d^*$ be the solution of $\arg\max_q \mathcal{J}_\beta(q,\pi)$. Then*

$$q_r^*(r|s,a) \propto p(r|s,a) \exp\left(\frac{r}{\beta}\right), \quad q_d^*(s'|s,a) \propto p(s'|s,a) \exp\left(V_\pi^*(s')\right). \tag{25}$$

*Proof.* We have that

$$\mathcal{J}_\beta(q^*,\pi) = V_\pi^*(s) = \max_{q_\pi} \mathbb{E}_{q_\pi}\left[\frac{r}{\beta} - \log\frac{q_r(r|s,a)}{p(r|s,a)} + V_\pi^*(s') - \log\frac{q_d(s'|s,a)}{p(s'|s,a)}\right]$$

$$= \mathbb{E}_{a\sim\pi}\left[\max_{q_r\in\Delta_\mathcal{R}} \mathbb{E}_{r\sim q_r}\left[\frac{r}{\beta} - \log\frac{q_r(r|s,a)}{p(r|s,a)}\right] + \max_{q_d\in\Delta_\mathcal{S}} \mathbb{E}_{s'\sim q_d}\left[V_\pi^*(s') - \log\frac{q_d(s'|s,a)}{p(s'|s,a)}\right]\right].$$

where the second equality comes from the definition of $V_\pi^*$. Using Lemma 4, we conclude that

$$q_r^*(r|s,a) \propto p(r|s,a) \exp\left(\frac{r}{\beta}\right), \quad q_d^*(s'|s,a) \propto p(s'|s,a) \exp\left(V_\pi^*(s')\right).$$

□

# B RISK-AVERSE M-STEP

In this section, we derive Eq. (11) for the two cases: $\beta > 0$ and $\beta < 0$. For $\beta > 0$, we have that the M-step:

$$\arg\max_\pi \mathcal{J}_\beta(q^*,\pi) = \arg\max_\pi \mathbb{E}_{q_\pi^*(\tau)}\left[\sum_t \frac{r_t}{\beta} - \log\frac{q_d^*(s_{t+1}|s_t,a_t)}{p(s_{t+1}|s_t,a_t)} - \log\frac{q_r^*(r_t|s_t,a_t)}{p(r_t|s_t,a_t)}\right]$$

$$= \arg\max_\pi \mathbb{E}_{q_\pi(\tau)}\left[\sum_t r_t - \beta\log\frac{q_d^*(s_{t+1}|s_t,a_t)}{p(s_{t+1}|s_t,a_t)} - \beta\log\frac{q_r^*(r_t|s_t,a_t)}{p(r_t|s_t,a_t)}\right].$$

where the second equality comes from multiplying by $\beta$. For $\beta < 0$, we have that the M-step:

$$\arg\min_\pi \mathcal{J}_\beta(q^*,\pi) = \arg\min_\pi \mathbb{E}_{q_\pi^*(\tau)}\left[\sum_t \frac{r_t}{\beta} - \log\frac{q_d^*(s_{t+1}|s_t,a_t)}{p(s_{t+1}|s_t,a_t)} - \log\frac{q_r^*(r_t|s_t,a_t)}{p(r_t|s_t,a_t)}\right]$$

$$= \arg\max_\pi \mathbb{E}_{q_\pi(\tau)}\left[\sum_t r_t - \beta\log\frac{q_d^*(s_{t+1}|s_t,a_t)}{p(s_{t+1}|s_t,a_t)} - \beta\log\frac{q_r^*(r_t|s_t,a_t)}{p(r_t|s_t,a_t)}\right].$$

where the $\arg\min$ becomes $\arg\max$ in the second equality due to multiplying through by negative $\beta$. Hence, for both cases the M-step is equivalent to Eq. (11). This shows that for $\beta < 0$, the overall optimization is equivalent to the saddle-point problem: $\arg\min_\pi \arg\max_q \mathcal{J}_\beta(q,\pi)$. Thus, for the risk-averse setting we do not have monotonic improvement on the entropic objective. Nonetheless, it

serves as an approximation to the unconstrained objective when $q(\tau) = p(\tau|\mathcal{O}_{1:T} = 1)$, for which we have equality to the entropic objective (Osogami, 2012). This surrogate objective is also related to Robust MDPs (Nilim & El Ghaoui, 2005) by treating the maximization of $q$ as the worst choice for an uncertain set, i.e. the set of variational distributions of the form $q_\pi(\tau)$. Additionally, the objective is equivalent to the Minimax Criterion under inherent uncertainty (García & Fernández, 2015; Heger, 1994) when $\beta \to 0$.

## C  DUAL OPTIMIZATION

Choosing a suitable $\beta$ can be a deciding factor between learning risk-sensitive policies or divergence in practice. Hence, we propose a Langrangian formulation that automatically tunes the risk-sensitive parameter $\beta$ for, both, the risk-seeking ($\beta > 0$) and risk-averse ($\beta < 0$) settings. For $\beta > 0$, we observe that the maximization of $\mathcal{J}_\beta(q, \pi)$ w.r.t. $q$ reflects the Lagrangian of the following constrained optimization,

$$\max_q \mathbb{E}_{q_\pi(\tau)} \left[ \sum_t r_t \right] \text{ s.t. } \mathrm{KL}(q_\pi(\tau) \,\|\, p_\pi(\tau)) \leq \epsilon, \tag{26}$$

where $\epsilon$ sets a hard-constraint on the allowable divergence of the distribution $q_\pi(\tau)$. We recognize $\beta$ as a Lagrange multiplier and perform dual gradient descent (Boyd & Vandenberghe, 2004) via the loss function:

$$J(\beta) = \beta \left( \epsilon - \mathrm{KL}(q(s_{t+1}, r_t \mid s_t, a_t) \,\|\, p(s_{t+1}, r_t \mid s_t, a_t)) \right). \tag{27}$$

Although constraint in the primal problem of Eq. (26) suggests optimizing the dual parameter $\beta$ w.r.t. the entire trajectory $\mathrm{KL}(q_\pi(\tau) \,\|\, p_\pi(\tau))$. This can lead to high variance for long trajectories. We instead impose the constraint only at single transition, which yields more stable learning. This approach most closely aligns with SAC (Haarnoja et al., 2018), which introduces a dual relaxation to modulate its policy entropy. For $\beta < 0$, we now consider the following primal problem,

$$\max_q \mathbb{E}_{q_\pi(\tau)} \left[ \sum_t c_t \right] \text{ s.t. } \mathrm{KL}(q_\pi(\tau) \,\|\, p_\pi(\tau)) \leq \epsilon. \tag{28}$$

where the optimization is w.r.t. costs $c_t = -r_t$. In other words, the agent aims to find the worst-case dynamics $q$ that are within $\epsilon$ of the true dynamics in a KL sense. Now consider the dual problem with Lagrange multiplier $\lambda$:

$$\min_{\lambda > 0} \max_q \mathbb{E}_{q_\pi(\tau)} \left[ \sum_t c_t \right] + \lambda(\epsilon - \mathrm{KL}(q_\pi(\tau) \,\|\, p_\pi(\tau))). \tag{29}$$

In particular, we observe that for any fixed $\lambda$ the maximization w.r.t. $q$ is equivalent to maximizing $\mathcal{J}_\beta(q, \pi)$ with $\beta = -\lambda$. Hence, we propose a dual gradient descent optimization with loss function:

$$J(\lambda) = \lambda \left( \epsilon - \mathrm{KL}(q(s_{t+1}, r_t \mid s_t, a_t) \,\|\, p(s_{t+1}, r_t \mid s_t, a_t)) \right). \tag{30}$$

where we can recover $\beta$ by setting it to $-\lambda$. Again, we impose the constraint only at single transitions, which yields more stable learning.

## D  ADDITIONAL EXPERIMENTS

### D.1  VISUALIZATION OF VARIATIONAL DYNAMICS FOR RSVAC

In Fig. 8, we visualize the learned variational dynamics on the stochastic continuous 2D environment for a range of $\beta$ values. When $\beta < 0$, we observe that the variational dynamics model is pessimistic and moves the agent towards the horizontal sides of the square. In contrast, when $\beta > 0$ the variational dynamics guide the agent towards the regions of high reward and ignore the potential of hitting the walls.

### D.2  ABLATION EXPERIMENTS FOR RSVAC

Ablation experiments for rsVAC using SAC as its actor-critic w.r.t. risky MuJoCo benchmarks (InvertedPendulum, HalfCheetah and Swimmer) for a range of $\beta$ values. Fig. 8 demonstrates that rsVAC is capable of learning risk-sensitive policies in, both, the risk-averse and risk-seeking regimes while achieving high reward.

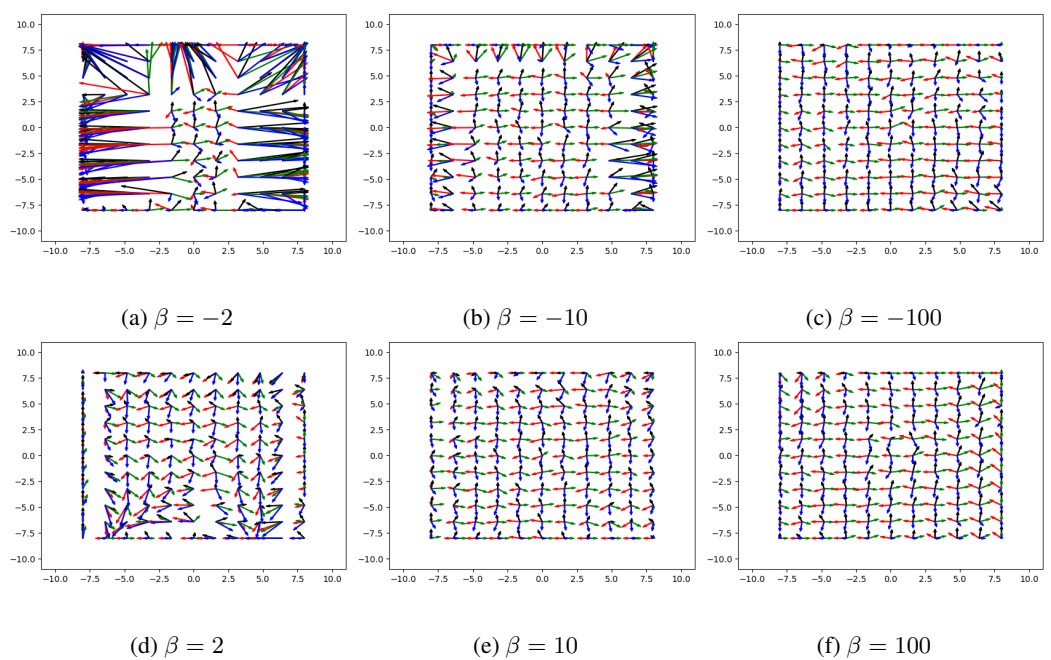

Figure 7: Visualizations of variational dynamics for linearly spaced coordinates in stochastic continuous 2D environment. From each state, we draw a vector to its expected next state colored by the agent's action: up (black), right (green), down (blue) and left (red).

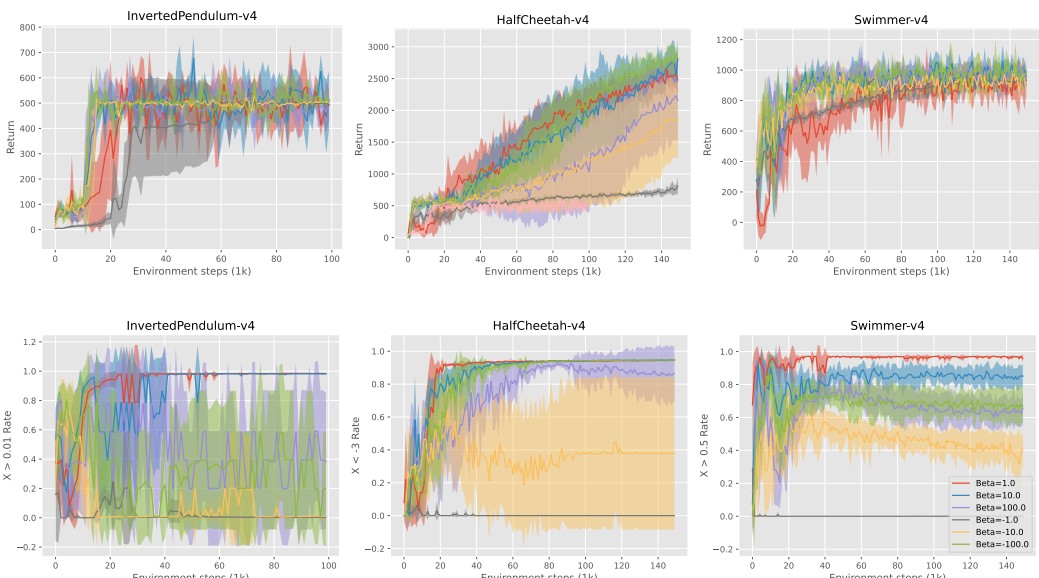

Figure 8: Ablation analysis w.r.t risk parameter $\beta$. The solid curves correspond to the mean and shaded regions to $\pm$ one standard deviation over 5 random trials. *Top row*: Average return. *Bottom row*: Percentage of steps on an episode in risky regions.

### D.3   ADDITIONAL MUJOCO EXPERIMENTS

Fig. 9 compares rsVAC (with TD3 as its actor-critic) to other risk-averse baselines on the MuJoCo environment Ant-v4. Similarly to our previous experiments, we modify this environment by including an additional stochastic reward sampled from $\mathcal{N}(0, 10^2)$ if X-position $> 0.5$. Again we observe that rsVAC is effective at learning risk-sensitivity while producing better policies in terms of average return.

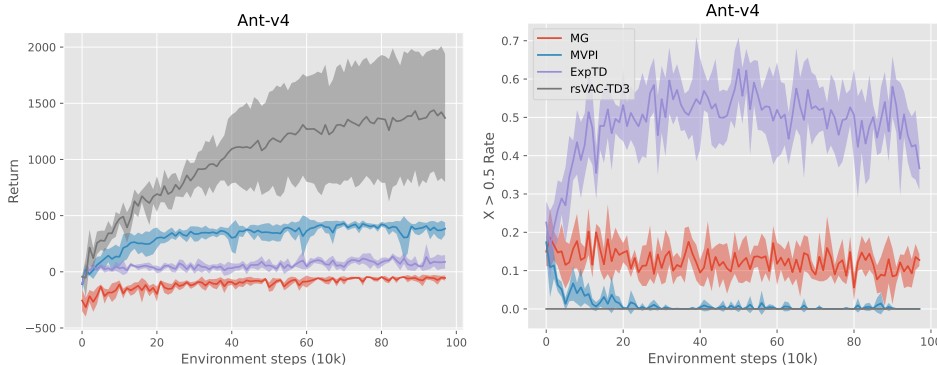

Figure 9: **Risky Ant-v4.** *Left*: Average return on Ant-v4. *Right*: Percentage of steps on an episode in risky regions. The solid curves correspond to the mean and shaded regions to $\pm$ one standard deviation over 5 random trials.

# E  IMPLEMENTATION DETAILS

## E.1  STABILITY MODIFICATIONS

During the implementation of rsVAC, we noticed that the log-terms in the critic update have no effects on controlling the risk-sensitivity of the algorithm, while producing instabilities that can hurt the critic's convergence. For this reason, we remove them during optimization of rsVAC for the continuous experiments. Another modification that we found that can improve learning for the variational dynamics is the introduction of a separate critic $V$ optimized w.r.t. real environment data. This critic is convenient as it provides information about the return in future states while remaining independent of the variational dynamics so it doesn't tend to become overly optimistic (or pessimistic) for $\beta$ values with small magnitude.

## E.2  CONTINUOUS 2D ENVIRONMENT

Table 1 lists the hyperparameters used by rsVAC for the stochastic continuous 2D environment.

Table 1: Hyperparameters for stochastic continuous 2D environment

| Schedule details | |
| --- | --- |
| Environment steps before training | 5000 steps |
| Environment steps per epoch | 1000 steps |
| Model optimization | every 1 steps |
| Number of model rollouts | 128 rollouts |
| Rollout length | 1 step |
| Network details | |
| Discount factor | 0.9 |
| Soft target update | 0.005 |
| Experience buffer $\mathcal{D}_{\mathrm{env}}$ | 1,000,000 |
| Model buffer $\mathcal{D}_{\mathrm{model}}$ | 128 |
| Dynamics Network Architecture | MLP with 2 hidden layers of size 256 |
| Actor Network Architecture | MLP with 2 hidden layers of size 256 |
| Critic Network Architecture | MLP with 2 hidden layers of size 256 |
| Network optimizer | Adam |
| Non-linear layers | ReLU |
| Learning rate | 0.0003 |

### E.3 MUJOCO ENVIRONMENTS HYPERPARAMETERS

For MG and MVPI, we use the implementations in Luo et al. (2024) and follow the same hyperparameters suggested by the authors. We implement expTD (Noorani et al., 2023) on top of the TD3 implementation in Luo et al. (2024) and select the -10 as its risky parameter from $\{-1, -10, -20, -100\}$. For rsVAC we select its risk parameter from $\{-1, -4, -8\}$. For SAC we use a re-implementation of that algorithm made available by other authors[2]. We use the same network architectures and learning rates for all algorithms. Table 2 lists the hyperparameters used for rsVAC and all actor-critic algorithms in risk-aware MuJoCo benchmarks.

Table 2: Hyperparameters for risk-aware MuJoCo benchmark

| Schedule details | |
|---|---|
| Environment steps before training | 5000 steps |
| Environment steps per epoch | 1000 steps |
| Model optimization | every 1 step |
| Number of model rollouts | 256 rollouts |
| Rollout length | 1 step |
| Network details | |
| Discount factor | 0.99 |
| Soft target update | 0.005 |
| Experience buffer $\mathcal{D}_{\text{env}}$ | 1,000,000 |
| Model buffer $\mathcal{D}_{\text{model}}$ | 256 |
| Reward Network Architecture | MLP with 2 hidden layers of size 256 |
| Actor Network Architecture | MLP with 2 hidden layers of size 256 |
| Critic Network Architecture | MLP with 2 hidden layers of size 256 |
| Network optimizer | Adam |
| Non-linear layers | ReLU |
| Learning rate | 0.0003 |
| $\beta$ initialization | -1 (-8 for invertedPendulum with TD3) |
| $\alpha$ initialization | 0.2 (for SAC implementation only) |

## F PSEUDOCODE OF RSVAC

This section contains the pseudocode for our algorithm, rsVAC.

---

[2]https://github.com/Xingyu-Lin/mbpo_pytorch

---

**Algorithm 1** rsVAC

---

Initialize networks, parameters and replay buffers.
**for** each epoch **do**
    **for** each environment step **do**
        $a_t \sim \pi_\theta(a_t|s_t)$
        $s_{t+1}, r_t \sim p(s_{t+1}, r_t|s_t, a_t)$            Sample next state from environment.
        $\mathcal{D}_{\text{env}} \leftarrow \mathcal{D}_{\text{env}} \cup \{(s_t, a_t, s_{t+1}, r_t)\}$         Add tuple to experience buffer.
        **if** model optimization **then**
            $\{(s_t^i, a_t^i, s_{t+1}^i, r_t^i)\}_{i=1}^N \sim \mathcal{D}_{\text{env}}$     Sample every tuple in experience buffer.
            $\theta \leftarrow \theta - \nabla J_d(\theta)$            Update prior dynamics $p_\theta$.
            $\theta \leftarrow \theta - \nabla J_r(\theta)$            Update prior reward $p_\theta$.
            $\phi \leftarrow \phi - \nabla J_d(\phi)$            Update variational dynamics $q_\phi$.
            $\phi \leftarrow \phi - \nabla J_r(\phi)$            Update variational reward $q_\phi$.
            **for** $m = 1, 2, ..., M$ **do**
                $s_t \sim \mathcal{D}_{\text{env}}$            Sample state from experience buffer $\mathcal{D}_{\text{env}}$.
                $a_t \sim \pi_\theta(a_t|s_t)$            Sample action using policy.
                $s_{t+1} \sim q_\phi(s_{t+1}|s_t, a_t)$    Sample next state using variational dynamics.
                $r_t \sim q_\phi(r_t|s_t, a_t)$        Sample reward using variational reward model.
                $\mathcal{D}_{\text{model}} \leftarrow \mathcal{D}_{\text{model}} \cup \{(s_t, a_t, s_{t+1}, r_t)\}$     Add tuple to model buffer.
            **end for**
        **end if**
        **for** $k = 1, 2, ..., K$ **do**
            $\{(s_t^i, a_t^i, s_{t+1}^i, r_t^i)\}_{i=1}^B \sim \mathcal{D}_{\text{model}}$    Sample mini-batch from model buffer $\mathcal{D}_{\text{model}}$.
            $\psi \leftarrow \psi - \nabla J(\psi)$            Update critic $Q_\psi$.
            $\theta \leftarrow \theta - \nabla J(\theta)$            Update policy $\pi_\theta$.
            $\psi' \leftarrow \tau\psi + (1 - \tau)\psi'$         Update target critic $Q_{\psi'}'$.
        **end for**
    **end for**
**end for**

---

