# OpenReview forum: "Risk-Sensitive Variational Actor-Critic: A Model-Based Approach"
_ICLR.cc/2025/Conference — ICLR 2025 Poster_

### Official Review · Reviewer_azQM · 2024-10-30

**Soundness:** 3
**Presentation:** 3
**Contribution:** 3
**Rating:** 8
**Confidence:** 3

**Summary:**

The paper presents a risk-sensitiv RL algorithm based on the idea of RL-as-inference. It presents a thorough derivation and a diverse set of experiments that highlights how the agent can be tuned to be risk seeking or avoiding with a simple parameter.

**Strengths:**

Overall, the paper is easy to follow, if somewhat dense in places. The idea is straightforward an well executed, building on a well-established framework.

The experimental section is thorough and contains both interesting small scale and reasonable large scale experiments. It would have been interesting to see if more complex or large dynamics systems such as image-based observations pose a challenge to the approach (I assume it would due to the difficulty of parameterizing a dynamics model?), but this is clearly something that can be relegated to future work.

Overall, my review is somewhat shorter than normal because I genuinely can't find too much to criticize. I think it's well done!

**Weaknesses:**

Overall, the paper does not have any major weaknesses that I would consider crucial for the decision to accept or reject.

As pointed out, the paper is somewhat dense in its writing. Some of the mathematical formalism could potentially moved to the appendix to declutter the main body, but given that the ideas presented are relatively standard and don't require exotic notation, this does not detract from the readability too much. The main thing contributing to this is that inline equations are used somewhat frequently.

One point that caused me a bit of confusion was that beta acts somewhat unintutively in that small absolute values of beta correspond to more extreme risk-averseness/seeking behavior, while large values correspond to the standard setting. This is briefly pointed out, but it could be elaborated on a bit more so that readers don't stumble.

**Questions:**

For standard RL as inference, it is known that variational algorithms produce optimistic policies if the transition dynamics are learned and optimized over. This is acknowledged in Section 3, yet I would encourage the authors to provide a more in-depth discussion and maybe some experiments if and when this poses a problem, especially for risk-averse agents? Also, is this phenomenon reversed for risk-averse agents, do they become more pessimistic? I expect some of this depends on the ratio of r_max vs beta?

---

> ### Author Response · Authors · 2024-11-22
>
> * **One point that caused me a bit of confusion was that beta acts somewhat unintutively in that small absolute values of beta correspond to more extreme risk-averseness/seeking behavior, while large values correspond to the standard setting. This is briefly pointed out, but it could be elaborated on a bit more so that readers don't stumble.**
>
> We thank the reviewer for their time and thoughtful feedback. We agree with the reviewer that this point could use extra clarification. This choice was made so that $\beta$ plays the role of a Lagrange multiplier in the dual optimization (Appendix C).  We have revised the manuscript with: "We emphasize that a risk-neutral policy is recovered for large $|\beta|$ values, while small $|\beta|$ values produce risk-seeking/averse policies."
>
> * **For standard RL as inference, it is known that variational algorithms produce optimistic policies if the transition dynamics are learned and optimized over. This is acknowledged in Section 3, yet I would encourage the authors to provide a more in-depth discussion and maybe some experiments if and when this poses a problem, especially for risk-averse agents?**
>
> We concur with the reviewer's comment. We have extended our discussion for Sec. 3: "Risk-sensitivity arises in Eq.(7) from the maximization w.r.t.~the variational distribution $q$. Although the penalty discourages deviations from the true model, the agent is willing to pay this penalty if the increase in expected return is large enough to compensate this extra cost. When $\beta > 0$, the variational model becomes optimistic (risk-seeking) as it aims to increase the expected return. When $\beta < 0$, it becomes pessimistic (risk-averse), as instead, it aims to increase the expected cost. Finally, we recover the true model when $|\beta| \to \infty$ as the objective only suffers the deviation penalty."
>
> * **Is this phenomenon reversed for risk-averse agents, do they become more pessimistic? I expect some of this depends on the ratio of r_max vs beta?**
>
> The reviewer's intuition is correct.  For risk-averse agents the dynamics are pessimistic --- the variational model aims to maximize expected cost and $\beta$ controls the allowed deviation from the true model, in a KL sense.

---

> > ### Comment · Reviewer_azQM · 2024-11-22
> > **Reviewer acknowledgement**
> >
> > Thanks for the reply. I am happy to keep my score.

---

### Official Review · Reviewer_m2Pi · 2024-10-31

**Soundness:** 3
**Presentation:** 2
**Contribution:** 2
**Rating:** 5
**Confidence:** 3

**Summary:**

This paper proposes to solve risk-sensitive reinforcement learning (RL) problems with entropic risk measures based on control as a inference perspective. After deriving the evidence lower bound (ELBO) of the log-marginal likelihood of optimal trajectory, the introduces a EM-based algorithm to maximize the ELBO. For E-step, the author introduces a Bellman-like operator and arguses that it is a contraction map. Based on this theoretical result, the E-step can be implemented as a successive application of this Bellman-like operator. M-step is introduced as a standard RL problem with augmented reward function. The proposed method can be interpreted as a model-based actor critic method. Experimental studies shows the effectiveness of the proposed method.

**Strengths:**

- Risk-sensitive RL is an important research direction and entropic risk measure is a popular risk measure. Practical algorithm for this objective has a large group of potential audiences (significance).
- Experimental results are informative. In particular, the experimental studies in Section 6.1 and 6.2 are very good show cases of the concept of this paper (quality, significance).

**Weaknesses:**

Despite the strengths, the paper has clear weaknesses.

### Major concerns

- I am not convinced of the validity of the theoretical results. The proof is not concrete and self-complete, and the assumptions are not clearly stated. As a result, I am also not convinced of the validity of the proposed algorithm, despite its empirical performance. The followings are my concerns on the theoretical arguments.
  - L.732, the distribution $p$ is different for $s^{\prime}$ and $r$ as we can see from Eqs. (18) and (19). How the second equality holds?
  - How the contraction statement in L.739 - L.743 follows? I believe the inequality in L.741 - L.742 is not trivial for the current definition of $\mathcal{T}_{\pi}$ and needs a formal proof.
  - How the successive application of $\mathcal{T}_{\pi}$ results in the second equality in L.755? In general, maximum and expectation is not commute.
  - Even though the main text does not assume that the horizon of MDP, $T$, is a random variable, "the transient assumption of stopping MDPs" is used in the beginning of the page 15.

- In the abstract, the author states that "algorithms that optimize this objective have been restricted to tabular or low-dimensional continuous environments". However, the experiments in Section 6.3 is still limited to the smaller problems in Mujoco environments. To better demonstrate the scalability of the proposed method, experimental evaluation in larger domain, such as Humanoid, is necessary.


### Minor issues

- L.752, typo: T -> \mathcal{T}_{\pi}


## After reading author responses and revised manuscripts
- Thank you for revising the paper. The theoretical arguments are almost clear. I raise the score from 3 to 5 (as of 11/25).

**Questions:**

### Suggestion
Please address my concerns, especially on the theoretical arguments. If the theoretical results are validated, I am happy to re-evaluate the manuscript.

---

> ### Author Response · Authors · 2024-11-22
>
> * **L.732, the distribution $p$ is different for $s'$ and $r$ as we can see from Eqs. (18) and (19). How the second equality holds?**
>
> We thank the reviewer for their time and for their thorough analysis of the theoretical aspects of our work.  The result of the second equality referenced comes from the assumed conditional independence of the reward and next state: $p(r,s’|s,a) = p(r|s,a)p(s’|s,a)$. We have updated the appendix to clarify this detail.
>
> * **How the contraction statement in L.739 - L.743 follows? I believe the inequality in L.741 - L.742 is not trivial for the current definition of and needs a formal proof.**
>
> For this result we demonstrated that our contraction operation is equivalent to the contraction operator proved in VMBPO (Chow et al., 2021).  We agree with the reviewer that a self-contained proof is preferable, but have been unable to reconstruct the proof of Chow et al. (2021) from first principles in the available timeline.  We have instead simplified the analysis to the finite horizon case for a self-contained proof. Please see Theorem 1 and the associated proof in the revised manuscript.
>
> * **In general, maximum and expectation is not commute.**
>
> The ability to commute the max and expectation operators is due to the Principle of Optimality and is a standard result of the optimal control literature (see Sec. 1.5 in Bertsekas Vol I, 2012).  Our revised analysis in the finite horizon setting also invokes this property analogous to Bertsekas' Prop. 1.3.1.
>
> * **Even though the main text does not assume that the horizon of MDP, $T$, is a random variable, "the transient assumption of stopping MDPs" is used in the beginning of the page 15.**
>
> Our revised finite horizon analysis no longer invokes the transient assumption of stopping MDPs.

---

> ### Comment · Reviewer_m2Pi · 2024-11-25
> **Thank you for revising; follow-up questions**
>
> I appreciate the authors' extensive effort to revise the paper. Since my concerns on theoretical arguments are largely addressed, I raised the score from 3 to 5. Could the authors leave the comment on the followings as well?
> - I am still confused why the equality holds at L732 from the Principle of Optimality (PO). I think it is not trivial that (19) satisfies PO in terms of variational distributions. Could you please elaborate and justify this claim more?
> - It seems that, one of my original concerns, "the experiments are still limited to the smaller problems", is not addressed. I understand that running additional experiments is not an easy thing, especially on large state-action environments. Could you justify that the current size of environments are enough for the scope of this paper? What kind of and size of applications do the authors have in mind?

---

> > ### Author Response · Authors · 2024-11-28
> >
> > * **I think it is not trivial that (19) satisfies PO in terms of variational distributions. Could you please elaborate and justify this claim more?**
> >
> > Thanks for requesting clarification on this point.  Proof of this result for the general DP setting can be found in Bertsekas (2012, Vol. 2, Appendix A).  For clarity we have included an extended discussion of this result for our specific setting in the appendix.
> >
> > * **One of my original concerns, "the experiments are still limited to the smaller problems", is not addressed. Could you justify that the current size of environments are enough for the scope of this paper? What kind of and size of applications do the authors have in mind?**
> >
> > Our initial claim about these algorithms being limited to low-dimensional environments came from the environments tested in RS-AC (Noorani et al.) where only inverted-pendulum and acrobat were considered. The main reason for the current size of our environments is to replicate the same benchmark set by Luo et al. which allows us to compare with a variety of baselines. However, our algorithm is also applicable to higher-dimensional environments. To demonstrate this, we have incorporated an extra experiment for the Ant environment (105 dimensions) in the appendix.

---

> ### Comment · Reviewer_m2Pi · 2024-12-03
> **Thank you for your response; follow-up question**
>
> I appreciate again the authors' extensive effort to add additional experiments and clarifications. The experimental result in Ant-v4 seems promising. I have two additional questions.
> - In order to assure that the assumption at __L759-L763__ holds, we need an assumption on the support of $q^{t}_d$, $q^{t}_r$, $p^{t}_d$ and $p^{t}_r$. Could you explicitly state the support assumption and, elaborate how and why this support assumption holds in the experiments?
> - Could you elaborate how the inequality __L781-786__ holds from __L771-L778__? The scalar $\epsilon$ is arbitrary but it needs to be positive. Thus, it seems that the r.h.s. of __L781-786__ does not lower bound the most r.h.s. of __L771-L778__.

---

> > ### Author Response · Authors · 2024-12-04
> >
> > * **In order to assure that the assumption at L759-L763 holds, we need an assumption on the support of $q_d^t$, $q_r^t$, $p_d^t$ and $p_r^t$. Could you explicitly state the support assumption and, elaborate how and why this support assumption holds in the experiments?**
> >
> > Regarding support, due to the presence of KL terms the model $p(\cdot)=0$ implies that the variational distributions $q(\cdot) = 0$ (absolute continuity).  Given that $Q_k(s_k,a_k)$ is bounded above by $\log E_{\pi}[\exp(\sum_{t=k}^T r_t/\beta)|s_k,a_k]$, then a sufficient condition for the assumption at L759-L763 to hold is for the $\log E[\exp(\cdot)]$ term to be finite. In particular, this term is finite when rewards are bounded or when the rewards are Gaussian with bounded means, which holds for our experiments.  We will make these clarifications in a final version of the paper.
> >
> > * **Could you elaborate how the inequality L781-786 holds from L771-L778? The scalar $\epsilon$ is arbitrary but it needs to be positive. Thus, it seems that the r.h.s. of L781-786 does not lower bound the most r.h.s. of L771-L778.**
> >
> > The validity of the bound in question is a standard analysis result.  Assume $a\geq b-\epsilon$ for all $\epsilon > 0$. Now suppose $a<b$ then we could set $\epsilon = \frac{b-a}{2} > 0$. Thus, $a \geq b - \frac{b-a}{2} = \frac{b+a}{2}$ from which we obtain that $a\geq b$ which is a contradiction. Hence, it must be that $a \geq b$.
> >
> > We thank the reviewer for the points they have raised, all of which we feel have strengthened the submission.  As the discussion period is at a close we kindly ask that you please consider whether our responses have addressed your remaining concerns.  If your concerns have been addressed we would be truly grateful if you would reconsider your evaluation of the paper based on the clarification provided in our responses.  Thanks again for all of your feedback!

---

### Official Review · Reviewer_XK4d · 2024-11-02

**Soundness:** 2
**Presentation:** 2
**Contribution:** 2
**Rating:** 5
**Confidence:** 4

**Summary:**

This paper proposes a risk sensitive model based actor critic algorithm based on variational inference principles; in a risk aware RL setting. The authors extend the setting to stochastic rewards, compared to past works on deterministic reward setting when studying risk sensitive RL, and proposes a EM style algorithm that alternates between learning the model and optimizing the policy. The algorithm is derived based on RL as probabilistic inference; studies the risk awareness in policy optimization setting; studies the challenges of model learning in stochastic reward setting - and experimentally demonstrates the efficacy of the proposed algorithm on a suite of benchmarks.

**Strengths:**

This paper has some of the following strengths :

1. Unlike past works, the authors study both the risk-seeking and risk-averse settings and proposes a model based policy optimization algorithm based on first principles of RL as inference.
2. The algorithm is derived based on standard formulation of entropic risk measure and the proposed objectives are derived based on the risk criterion.
3. The paper is well written with the derivation of the key objectives for learning the model (rewards and transitions) and policy optimization well explained.
4. Overall, for this setting, there are few aspects to consider when learning both the model and the policy. The authors extend the setting to the stochastic rewards setting and shows how a suitable model can be learnt even under stochastic rewards - a common setting to consider when studying risk awareness of learnt policies in RL.
5. The EM Style algorithm derivation and the alternative optimization of the objectives are natural to consider when studying in this setting - and the authors do a good job in deriving a suitable algorithm that can work empirically.
6. Experiment results demonstrates the usefulness of the rsVAC algorithm under few tasks - and takes some standard baselines on top of which the risk awareness criterion can be easily integrated to study effectiveness of the proposed algorithm.

**Weaknesses:**

Although the work seems good, there are few weaknesses that concern me about this work :

1. The alternative EM style optimization of the objectives seems to bring in a lot of instability in the overall learning process. This has been a common issue in several past works on risk aware RL; for example, past works that even studied mean-variance RL, or when minimizing a variance measure in presence of inherent stochasticity in the environment. Similar to past works - I am worried about the effectiveness of the algorithm, beyond tasks that are demonstrated here, with the minimal baselines that are used for comparison.

2. Experimental results do not seem convincing enough, and it is not clear what are the different aims of each experiment. It would be useful to better structure the experiments section to make it clear what different aspects the rsVAC algorithm seems promising in?

3. There are several baselines for comparison in this line of work. I understand the work extends it to stochastic rewards setting -but to be more comprehensive, it would be useful to even show how effective past algorithms are in the stochastic rewards setting?

4. I am not sure if the chosen baselines for experiments are suitable for the tasks? Can the authors compare to some more baselines?

5. The derivation of the algorithm, based on the EM style approach seems useful - but there has been past works which would do in a similar way? What’s the novelty here?

**Questions:**

1. Can the authors comment on the instability issues related to comment in weakness (1)?

2. Have there been any ablation study related to instability when optimizing the different objectives, involved in model learning and then doing policy optimization?

3. I would like to see more experiments comparing to past baselines studying risk aware RL; even if in deterministic settings. How well do those algorithms scale to stochastic rewards, and this can help demonstrate the effectiveness of rsVAC itself.

4. Are there simple suite of tasks, not necessarily control based Mujoco ones - where rsVAC can be shown to be substantially useful? Or I do not see the novelty in the algorithm - and the way it is derived is quite standard too; so maybe I am missing something here? Section 6.2 seems useful in this context - but what happens to policy learning in this setting?

5. Objective-wise, can the authors comment on what makes the proposed objectives interesting to look into, which past works have ignored?

---

> ### Author Response · Authors · 2024-11-22
>
> * **...there has been past works which would do in a similar way? What’s the novelty here?**
>
> We thank the reviewer for their time and useful insights.  EM-style algorithms have been proposed to optimize w.r.t. policies and dynamics models. We improve upon existing work in several ways.  Ours is the first variational approach that supports a stochastic reward model, and the first such approach to address risk-averse policy learning.  We additionally provide a rigorous formulation of the variational lower bound on the entropic objective, along with a thorough theoretical analysis on the Bellman-style operator.
>
> * **I would like to see more experiments comparing to past baselines studying risk aware RL...How well do those algorithms scale to stochastic rewards**
>
> We have conducted experiments with an extensive set of baselines.  If the reviewer has a specific baseline they would like to see comparison with please let us know and we can run it during the discussion phase.
>
> * **Are there simple suite of tasks...where rsVAC can be shown to be substantially useful?**
>
> There does not exist a standard suite of risk-sensitive learning tasks that we are aware of.  We have chosen tasks from existing literature that considers risk-sensitivity.  The tabular environment is adapted from Eysenbach et al. (2022) and demonstrates that rsVAC performs as well or better than baseline methods in tabular settings with risk induced by stochastic dynamics.  The stochastic 2D environment is of our own devising and demonstrates risk sensitivity in a continuous environment where function approximation is necessary.  The Mujoco experiments are adapted from contemporary risk-sensitive RL literature, specifically Luo et al. (2024).  This last set of experiments demonstrates that rsVAC is as good or better at learning risk-sensitive policies than SOTA baselines where risk arises due to stochastic rewards.
>
> * **Objective-wise, can the authors comment on what makes the proposed objectives interesting to look into, which past works have ignored?**
>
> Previous approaches cannot learn risk-averse policies, which are arguably more practical than risk-seeking policies in many settings (e.g. Safe RL). Additionally, ours is the first variational approach to our knowledge that explicitly models a stochastic reward. Hence, it can learn risk-sensitive policies even when the only source of risk comes from the rewards. Approaches like MPO and VMBPO ignore this type of reward risk.
>
> * **I am not sure if the chosen baselines for experiments are suitable for the tasks?**
>
> We compare to the best baselines that were considered in the work that proposed these stochastic tasks: MVPI, a risk-averse algorithm that aims to perform comparable to risk-neutral algorithms, and Gini measure, the work the proposed these stochastic tasks. Additionally, we consider a baseline that directly optimizes entropic risk measure (expTD) which is the same measure our algorithm approximately optimizes.  We can consider additional baselines if the reviewer can suggest specific comparisons.
>
> * **The alternative EM style optimization of the objectives seems to bring in a lot of instability in the overall learning process.**
>
> Our method can be affected by instabilities when working with small magnitude $\beta$ values. However, other methods that optimize the entropic-risk measure (Noorani et al., 2023) are susceptible to significantly more instabilities by having to compute exponentiated returns. Instability in expTD for example can be seen in the high return variance of Fig. 6, as opposed to the comparatively low variance of rsVAC.  By using the ELBO, our method avoids this numerical instability. Additionally, we explored mechanisms to alleviate the EM-style optimization instabilities such as introducing a separate critic V optimized w.r.t. real environment data.

---

> > ### Author Response · Authors · 2024-12-03
> > **Follow-up**
> >
> > Dear Reviewer,
> >
> > We thank you again for taking the time to review our work, and provide valuable insights.  We are hopeful that our rebuttal addresses the concerns you raised in your initial review.
> >
> > As the discussion period draws to a close we kindly ask that you please consider whether our responses have addressed your concerns.
> >
> > If your concerns have been addressed we would be truly grateful if you would reconsider your initial evaluation of the paper based on the clarification provided in our responses.
> >
> > Thanks again!
> > Authors

---

### Official Review · Reviewer_AxU6 · 2024-11-03

**Soundness:** 4
**Presentation:** 4
**Contribution:** 3
**Rating:** 8
**Confidence:** 3

**Summary:**

This work presents an approach to optimize the entropic risk objective in RL by formulating it as a RL-as-probabilistic-inference objective, and optimizing a variational lower bound. This lower bound is solved using two iterative steps, where in one step the variational dynamics of the environment are learned, and in the other step, an actor-critic algorithm optimizes for a policy with a modified reward.
The risk-sensitivity of the approach arises from the sign of a parameter $\beta$, and the notion of risk here is related to the variance of the returns. Experiments in tabular and continuous environments demonstrate that the algorithm can learn risk-seeking and risk-averse behaviors based on $\beta$ and outperforms similar algorithms in the risk-averse setting.

**Strengths:**

The presentation of the paper was mostly clear, rigorous, and well-written.
The illustrative example of Figure 1 was clear and helpful in understanding. There were a good variety of experiments in different domains. The 2D stochastic environment experiments nicely show the effect of using $\beta$ to modulate risk behavior of the policy.

The paper has originality and significance. From my understanding, these come from extending the RL-as-inference framework to stochastic rewards, and using this framework to also optimize for risk-averse policies.

**Weaknesses:**

- I think the presentation of the ELBO objective needs to be improved to clearly indicate that this is not an original objective of this paper. There are also many similarities to [Chow et al, 2021] in the overall formulation and the Bellman-like operators in the E-step optimization. The difference between these works needs to be highlighted more than what is given in L302-303.
  - Since a major difference between the two works is given as VMBPO only optimizing for risk-seeking behavior, I am wondering: would you not be able to get a risk-averse policy by choosing the temperature $\eta < 0$ in VMBPO as well?


- The algorithm presented, rsVAC, is quite complicated and potentially subject to a lot of instabilities in practice. My main concerns are:
  - (E-Step, section 4.1) Learning the variational dynamics and reward requires first to learn a model of the true dynamics and reward, $p_\theta$. How does the error in $p_\theta$ affect the variational distributions?
  - $V_\psi$ is used in Eq. (15) to learn the variational dynamics, and on the other hand it is optimized in the actor-critic step using the learned variational dynamics. I expect this would be quite difficult to stabilize in practice. Could you comment on that?
  - $\beta$ parameter sensitivity: Is the convergence of the algorithm very dependent on the $\beta$ chosen? How was this sensitivity observed in the experiments?
- I don’t agree that Fig 3a shows rsVAC is more efficient at learning and obtains a higher return (L360). It looks like all the curves are within each other’s variability. I think more runs (~30) are needed before a conclusion can be made, which I feel is not unreasonable since this is a tabular environment.

Overall I think the originality and contribution of the paper are significant enough to be accepted, however, I have doubts as to the usability of the main algorithm in practice.

Minor comments:
- I found the presentation of (12) confusing, it would be better if they are presented as equations.

**Questions:**

- You mention in L040 that optimizing the entropic risk requires knowledge of the transition kernel, citing Noorani et al, 2023. To my knowledge the algorithms proposed in that work (particularly R-AC -- Risk-sensitive online Actor-Critic) do not require knowledge of the transition kernel and propose an actor-critic algorithm to optimize the entropic risk much like this work. Could you expand on the comparison between the two works and what advantage this variational approach has to the R-AC of [Noorani et al., 2023]?
- Regarding objective (7), you mention that the controller may still learn risk-seeking or risk-averse behavior in L160 “if the change in return sufficiently compensates”. What does this mean?
- In Section 4, it is mentioned that the model learning objectives are optimized using SGD. Since this is an online algorithm and these steps are done iteratively, does this mean that one step of SGD was taken for the optimization at each environment collection step or multiple steps, or optimized up to some convergence criteria? (e.g. L214, L222, L233) How was this decided in practice?


## Post-rebuttal
Increased my score as my questions have been answered

---

> ### Author Response · Authors · 2024-11-22
>
> * **You mention in L040 that optimizing the entropic risk requires knowledge of the transition kernel, citing Noorani et al, 2023. To my knowledge the algorithms proposed in that work...do not require knowledge of the transition kernel**
>
> We thank the reviewer for their time and helpful feedback.  The reviewer is correct that Noorani et al. does not require knowledge of the transition kernel.  We have updated the text to make this clear: "Directly optimizing this objective is challenging: it requires the knowledge of the transition kernel or it depends on unstable updates w.r.t. exponential Bellman equations (Noorani et al., 2023)."
>
> * **What advantage this variational approach has to the R-AC of [Noorani et al., 2023]?**
>
> R-AC updates are numerically unstable since the value functions compute exponentiated return. These instabilities can be seen in the high return variance of Fig. 6.  In practice one must set $\beta$ very large to avoid numerical overflow.  By using the ELBO, our method avoids the exponential operation which allows us to learn policies in higher-return environments and use smaller magnitude values for $\beta$.
>
> * **...you mention that the controller may still learn risk-seeking or risk-averse behavior in L160 “if the change in return sufficiently compensates”. What does this mean?**
>
> Although the penalty discourages deviations from the true model, the agent is willing to pay this penalty if the increase in expected return is large enough to compensate this extra cost.
>
> * **Since this is an online algorithm...does this mean that one step of SGD was taken for the optimization at each environment collection step or multiple steps, or optimized up to some convergence criteria?**
>
> In model-based RL it is standard to update the model up to convergence w.r.t. the collected data as this is essentially a supervised-learning task. However, our model depends on the learned dynamics and value functions to be accurate before the variational distribution can learn risk-sensitive dynamics/reward models. For this reason, we perform one SGD update for each step in the environment (identical to the actor-critic architecture updates).
>
> * **Presentation of the ELBO objective needs to be improved...There are also many similarities to [Chow et al, 2021]**
>
> The reviewer is correct that there are many similarities between our formulation and that of Chow et al. (2021), but there are also key differences.  The main differences are that the VMBPO approach only considers risk-seeking policies and deterministic rewards.  Our formulation extends this to risk-averse policies and accounts for stochastic rewards.  Another key difference is that VMBPO maintains a posterior policy ($q_c$ in their notation) that differs from the prior policy ($\pi$).  By constraining the variational policy to equal $\pi$ we ensure a valid M-step in the risk-averse setting (see response below).
>
> * **Would you not be able to get a risk-averse policy by choosing the temperature $\eta < 0$ in VMBPO as well?**
>
> One key distinction between VMBPO and our work is that VMBPO learns a variational policy ($q_c$). For this reason when $\eta < 0$, their M-step is equivalent to maximizing the KL between their variational policy and prior policy $\pi$, which is trivially infinity. In our case, we constrain the variational policy to equal $\pi$ and obtain a valid M-step for both the risk-averse and risk-seeking settings.
>
> * **(E-Step, section 4.1) How does the error in $p_\theta$ affect the variational distributions?**
>
> We found that at the beginning of learning the critic output is too small to influence the variational distributions. Hence, they tend to be close to the learned dynamics and reward model. Later in learning, if the learned true models have high variance that can result in very optimistic/pessimistic variational models. For this reason, we clipped the output variance of the learned true models.
>
> * **$V_\psi$ is used in Eq. (15) to learn the variational dynamics...I expect this would be quite difficult to stabilize in practice.**
>
> In practice, the variational dynamics can become overly optimistic (or pessimistic) for $\beta$ values with small magnitude which results in policies that are not useful. We found that learning can be stabilized by the introduction of a separate critic V optimized w.r.t. real environment data. This critic has the benefit that it is independent of the variational dynamics, therefore, it avoids the instabilities from the learned dynamics.

---

> > ### Author Response · Authors · 2024-11-22
> >
> > * **$\beta$ parameter sensitivity: Is the convergence of the algorithm very dependent on the chosen $\beta$?**
> >
> > We found that the convergence of our method was not overly dependent on the chosen parameter $\beta$ in the ablation experiments. However, the risk-sensitive behavior would disappear if not set appropriately. In addition we introduce a dual-optimization procedure for learning $\beta$ online (Appendix C) and find that learning tends to be insensitive to initialization (Fig. 3(b) and (c)).
> >
> > * **I found the presentation of (12) confusing, it would be better if they are presented as equations.**
> >
> > We thank the reviewer for noticing this confusing fact. We have incorporated the equations in the latest version.
> >
> > * **I don’t agree that Fig 3a shows rsVAC is more efficient at learning and obtains a higher return (L360). It looks like all the curves are within each other’s variability.**
> >
> > We agree with the assessment by the reviewer. We have updated the text: "rsVAC ($\beta > 0$) is as efficient and and performs as well or better than VMBPO and Q-learning".

---

> > > ### Comment · Reviewer_AxU6 · 2024-11-26
> > >
> > > Thank you for answering my questions. I believe all my concerns have been adequately answered, and in particular, the contributions with respect to VMBPO and R-AC have been clarified. I will therefore increase my score.

---

### Meta-Review · Area_Chair_ruoe · 2024-12-19

**Metareview:**

The paper studied risk-sensitive model based RL based on variational inference principles. Leveraging the RL-as-probabilistic-inference framework, the paper designs an ELBO loss which the authors claims is the key to the stability of their algorithm when compared to prior risk-sensitive approaches.

Reviewers raised concerns on the instability of the algorithm, not enough baselines, and the scalability of the approaches. The authors addressed these concerns during their rebuttal.

**Additional Comments On Reviewer Discussion:**

The two negative reviewers raised concerns on the proof, instability of the algorithm, not enough baselines, and the scalability of the approach. During the rebuttal, the authors addressed the concerns on the proof, argued that the proposed approach is actually more stable than prior work due to its usage of a ELBO loss. The authors also included additional experiments on a higher dimentional control problem. Together i think this addressed most of the concerns from the two negative reviewers.

---

### Decision · Program_Chairs · 2025-01-22

Accept (Poster)